**The impact of recent changes in Asian anthropogenic emissions of SO$_2$ on sulfate loading**
**in the upper troposphere and lower stratosphere and the associated radiative changes**
Suvarna Fadnavis[1], Rolf Müller[2], Gayatry Kalita[1], Matthew Rowlinson[2], Alexandru Rap[2], Jui-
Lin Frank Li[3], Blaž Gasparini[4] Anton Laakso[5]
[1]Indian Institute of Tropical meteorology, Pune, India
[2]Forschungszentrum Jülich GmbH, IEK7, Jülich, Germany
[3]School of Earth and Environment, University of Leeds, Leeds, UK.
[4]Jet Propulsion Laboratory, California Institute of Technology, Pasadena, California, USA
[5]Department of Atmospheric Sciences, University of Washington, Seattle, USA
[6]Finnish Meteorological Institute, Finland
Corresponding author: suvarna@tropmet.res.in
Abstract:
Convective transport plays a key role in aerosol enhancement in the upper troposphere
and lower stratosphere (UTLS) over the Asian monsoon region where low-level convective
instability persists throughout the year. We use the state of art ECHAM6–HAMMOZ global
chemistry-climate model to investigate the seasonal transport of anthropogenic Asian sulfate
aerosols and their impact on the UTLS. Sensitivity simulations for SO$_2$ emission perturbation
over India (48 % increase) and China (70 % decrease) are performed based on the Ozone
Monitoring Instrument (OMI) satellite observed trend; rising over India by ~4.8 % per year
and decreasing over China by ~ 7.0 % per year during 2006 – 2017. The enhanced Indian
emissions result in an increase in Aerosol Optical Depth (AOD) loading in the UTLS by 0.61
to 4.17 % over India. These aerosols are transported to the Arctic during all seasons by the
lower branch of the Brewer-Dobson circulation enhancing AOD by 0.017 % to 4.8 %.
Interestingly, a reduction of $SO_2$ emission over China inhibits the transport of Indian sulfate
aerosols to the Arctic in summer-monsoon and post-monsoon seasons due to subsidence over
northern India. The region of sulfate aerosols enhancement show significant warming in the
UTLS over North India, South China (0.2±0.15 to 0.8±0.72 K) and the Arctic (~1±0.62 to
1.6±1.07 K). The estimated seasonal mean direct radiative forcing at the top of the atmosphere
(TOA) induced by the increase in Indian $SO_2$ emission is -0.2 to -1.5 $W{\cdot}m^{-2}$ over northern
India. The Chinese $SO_2$ emission reduction leads to a positive radiative forcing of ~0.6 to 6
$W{\cdot}m^{-2}$ over China. The decrease in vertical velocity and the associated enhanced stability of
the upper troposphere in response to increased Indian $SO_2$ emissions will likely decrease
rainfall over India.
Keywords: sulfate aerosols, radiative forcing, upper troposphere, and lower stratosphere,
India, China.

1. **Introduction**

Emissions of sulfur dioxide ($SO_2$) were shown to have large detrimental effects on air quality, and therefore, human health. Moreover, increases in $SO_2$ have effects on the hydrological cycle and crop yield (Li et al., 2017; Shawki et al., 2018). On the other hand, $SO_2$ emissions have a cooling effect on climate, due to the increased formation of sulfate aerosols ($SO_4^{2-}$) which are produced from the oxidation of $SO_2$. Over the Asian region, the high emission growth of $SO_2$ also has implications on the recurrent and more severe droughts happening during the second half of the twentieth century resulting in socio-economic impacts (Kim et al., 2016; Paul et al., 2016; Zhang et al., 2012a). Its effects on precipitation deficit is via scattering of solar radiation leading to the invigoration of surface cooling, reduction in land-ocean thermal contrast, and overturning of circulation (Ramanathan et al., 2005, Yeh et al.,2015; Shawki et al., 2018).

To curb its adverse effect, implementation of international legislation on sulfur emission was enforced which resulted in global decrease until 2000 followed by a sharp rise until 2006 and declining trend afterward. The global rising and declining trend seem to be modulated by the emissions from China since it is the world largest $SO_2$ emitting country (Aas et al., 2019). While $SO_2$, emissions over China have declined since 2006 (by ~75%), India shows a continued increase (~50%) (Krotkov et al., 2016; Li et al., 2017). The rising trend in $SO_2$ emissions in India is due to sustained economic growth during the last few decades (Krotkov et al., 2016). According to the Indian Ocean Experiment (INDOEX) during January to March 1999 sulfate aerosols over the Indian region contribute 29 % to the observed aerosol optical depth (AOD) (Verma et al., 2012). The Aerosol Radiative Forcing over India NETwork (ARFINET) AOD measurements over India show a consistent rising annual trend of

0.004 during 1988 – 2013 (Babu et al., 2013).  Over North India sulfate AOD estimates vary
between ~ 0.10 and 0.14, and the direct radiative forcing (DRF) at TOA between ~ -1.25 to
and -2.0 W$\cdot$m$^{-2}$ (Verma et al., 2012). Globally, the current best estimate of sulfate aerosol
DRF is -0.4 W$\cdot$m$^{-2}$ (-0.6 W$\cdot$m$^{-2}$ to -0.2 W$\cdot$m$^{-2}$) (Myhre et al., 2013).
The long-range transport of sulfate aerosols from the Asian boundary layer to the UTLS
and further northward to the Arctic (poleward of 65 ⁰N) alter the aerosol burden in the upper
troposphere over Asia and the Arctic (Bourgeois and Bey, 2011; Yang et al., 2018). This
northward extending layer from Asia to the Arctic in the UTLS affects the surface temperature
and produces climatic impacts via DRF (Yang et al., 2018). The Cloud-Aerosol Lidar with
Orthogonal Polarization (CALIOP) satellite measurements and model simulations indicate that
13 % (annual mean) of the sulfate in the Arctic troposphere comes from Asia (Bourgeois and
Bey, 2011). The model sensitivity experiments for 20 % emission reduction of $SO_2$ show a
decrease in the sulfate aerosol burden in the Arctic by ~36 – 41 % when tagged with East
Asian emission and ~7 – 10 % in response to South Asian emissions. The global burden of
sulfate aerosols during 1975 – 2000 has produced a cooling trend of 0.02 K decade$^{-1}$ in
surface temperature (Yang et al., 2018). The recent significant changes in $SO_2$ emissions
within Asia are likely to alter the atmospheric burden of sulfate aerosols and their impacts (on
radiative forcing, clouds, temperature etc.), both regionally and at the remote locations.
The transport of aerosols from the Asian boundary layer to the UTLS by the monsoon
convection is known to form and maintain the Asian Tropopause Aerosols Layer (ATAL)
(SPARC-ASAP, 2006; Fadnavis et al., 2013; Vernier et al., 2015; Yu et al., 2017; Vernier et
al., 2018). In the future, the aerosol burden in the UTLS may increase due to rising trends in
aerosol emission. The enhancement in the UTLS involves complexities due to transport
processes. Previous work indicates that a fraction of Asian emissions is transported to the
UTLS (contributing to the ATAL associated with the monsoon anticyclone) since the majority
of aerosols that grow into cloud droplets (~80 %) is removed by precipitation. Two-thirds of
the total aerosol loading that reach the monsoon anticyclone is transported poleward through
circulation in the lower stratosphere (Lelieveld et al., 2018). The observed $SO_2$ concentrations
in the monsoon anticyclone are ~5 – 10 times higher than in the rest of the tropics (Lelieveld et
al., 2018). The major sources of aerosols in the ATAL are found in India and China, with
Indian emissions dominating the composition of the ATAL (Lau et al., 2018). Climate model
simulations show that the Asian monsoon region (15 – 45 °N, 30 – 120 °E) is three times more
efficient (per unit area and time) in enhancing aerosol in the Northern Hemisphere stratosphere
than annually - averaged tropical (15 °N – 15 °S) upwelling (Yu et al., 2017). Although the
chemical composition of the particles constituting the ATAL is not well understood, satellite
observations (e.g. Cloud-Aerosol Lidar and Infrared Pathfinder Satellite Observation,
CALIPSO;  Stratospheric Aerosol and Gas Experiment, SAGE–II; balloonsonde and aircraft
measurements (e.g. Civil Aircraft for the Regular Investigation of the atmosphere Based on an
Instrumented Container; CARIBIC) suggest that ATAL particles may contain large amounts
of sulfate, as well as black carbon, organic, nitrates (including ammonium nitrate) and dust
(Vernier et al., 2015; 2018; Yu et al., 2016; Höpfner et al., 2019). Further, model studies
suggest sulfate is, together with organics, a major chemical component of the ATAL (e.g.,
Fadnavis et al., 2013; Yu et al., 2017). However, there is also a model study (Gu et al., 2016)
that emphasizes the importance of nitrate as a chemical component of the aerosol in the UTLS
over the Tibetan Plateau and the South Asian summer monsoon region. In addition, balloon
measurements from Hyderabad, India indicate the presence of large amounts of nitrate
aerosols near the tropopause (100 ng m$^{-3}$), which may be due to $NO_X$ from anthropogenic
emissions, lightning, and gas-to-particle conversion (Vernier et al., 2015; 2018). Further, Yu et
al. (2016, 2017) report that sulfate and nitrate aerosols are important components of the
ATAL. Aerosol loadings in the UTLS result in a significant impact on radiative forcing. For
example, satellite observations show that the ATAL layer has exerted a regional radiative
forcing at the top of the atmosphere of approximately -0.1 $W \cdot m^{-2}$ in the past 18 years, thus
locally reducing the impact of global warming (Vernier et al., 2015).
Over Asia, the intensity of seasonal convection is controlled by regional instability and
thereby modulating the horizontal and vertical transport processes (Luo et al., 2013). The
transport pathways of pollutants lifted into upper troposphere by the monsoon convection are
well documented: (i) quasi-isentropic transport in the monsoon anticyclone above about 360 K
from the monsoon anticyclone into the extra-tropical lowermost stratosphere, (ii) cross-
isentropic transport from the UTLS into the tropical stratosphere by slow, radiatively driven
ascent, and (iii) transport of air into the stratosphere by deep convection that sometimes
crosses the tropopause in the tropics (Kremser et al., 2016; Fadnavis et al., 2017a; Vogel et al.,
2019). However little is known about the transport of Asian pollutants in the UTLS outside of
the summer monsoon.
In this study, we address the following research questions: (1) what is the seasonal
contribution of $SO_2$ emissions from India and China to the AOD in the UTLS? (2) what is the
associated radiative forcing? (3) can the increase/decrease in Indian/Chinese $SO_2$ emissions
change the seasonal dynamics and clouds in the UTLS? For this purpose, we perform two sets
of sensitivity simulations based on observed satellite trends in $SO_2$ emissions over India (48 %
increase) and China (70 % decrease) during 2006 - 2017 using the state of art aerosol-
chemistry-climate model ECHAM6–HAMMOZ (version echam6.1.0-ham2.1-moz0.8).
The paper is organized as follows: Section 2 describes the model simulations and
measurements used in our study. The model evaluation follows in Section 3. The distribution
of aerosols in the UTLS is discussed in Section 4. The impact of sulfate aerosols on radiative
forcing, cloud ice, and temperature are presented in Section 5. Discussions are given in section
6. Finally, section 7 presents the conclusions of this study.

**2. Measurements and model simulations**


**2.1 Satellite and ground-based measurements of AOD**


We analyze aerosol retrievals from Multi-Angle Imaging Spectroradiometer (MISR)
(level-3 version 4, at 550 nm wavelength during 2000 – 2016) (Martonchik et al., 2002), The
MISR AOD measurements give aerosol properties over the global ocean and land with bright
targets such as deserts (Kahn et al., 2001). Aerosol-Robotic-NETwork (AERONET) sun
photometer, level 2.0 version 3 daily AOD observations during 2006 – 2016 (Holben et al.,
1998) were also analyzed at the stations in the Indo–Gangetic Plain, (Bihar: 84.12 °E, 25.87
°N, Jaipur: 75.80 °E, 26.90 °N, Kanpur: 80.23 °N, 26.51 °N, Karachi: 67.13 °N, 24.95 °N),
and China (Xiang He: 39.76 °N, 11.00 °E, Nghia Do: 21.04°N, 105.80 °E).
**2.2    SO$_2$ measurements from the Ozone Monitoring Instrument (OMI)**
The Ozone Monitoring Instrument (OMI) aboard the NASA Aura spacecraft retrieves
SO$_2$ data from Earthshine radiances in the wavelength range of 310.5 – 340 nm (Levelt et al.,
2006). It gives the total number of $SO_2$ molecules in the entire atmospheric column above a
unit area (https://disc.gsfc.nasa.gov/datasets/OMSO2e_V003/). Details of the retrieval
technique are documented by Li et al., (2017). To understand the impact of $SO_2$ emission
changes over India and China, we estimate a trend in the $SO_2$ (2007 – 2017) over the Indian
region (70 – 95 °E, 8 – 35 °N) and the Chinese region (95 – 130 ºE; 20 – 45 ºN) (see Fig. 2e).
For this purpose, we used version 1.3, level-2, OMI retrievals that assume all $SO_2$ is located in
the planetary boundary layer. We use a regression model described by Fadnavis and Beig
(2006). A model regression equation is given as follows:
$\theta(t,z) = \alpha(z) + \beta(z) \, Dayindex \, (t)$ **(1)**
where $\theta(t,z)$ is the daily mean number of $SO_2$ molecules averaged over the Indian/Chinese
region, with altitude z set to 1 km, as we use column data. The model uses the harmonic
expansion to calculate the seasonal coefficient, $\alpha$, and the trend coefficient, $\beta$. The harmonic
expansion for $\alpha(t)$ is given as:
$\alpha(t) = A_0 + A_1 \cos \omega t + A_2 \sin \omega t + A_3 \cos 2\omega t + A_4 \sin 2\omega t$ **(2)**
Where $\omega = 2\pi/12$; $A_0$, $A_1$, $A_2$ ……. are constants and t (t=1,2 ….n) is the time index. The
estimated trend value for $SO_2$ is $4.8 \pm 3.2$ % $yr^{-1}$ over the Indian region and $7.0 \pm 6.3$ % $yr^{-1}$
over the Chinese region (99 % confidence interval). These trend values are used while
designing the model sensitivity simulations (discussed in section 2.4).


## 2.3 CloudSat and Cloud-Aerosol Lidar Infrared Pathfinder Satellite Observations (CALIPSO)

We use the ice water content (IWC) dataset from a combination of CALIPSO lidar and CloudSat radar data (2C–ICE dataset, version L3_V01) for the period 2007 – 2010 (Deng et al., 2013). The Cloud Profiling Radar (CPR) onboard the CloudSat satellite is a 94 GHz nadir-looking radar which measures the power backscattered by clouds as a function of distance. It provides information on cloud abundance, distribution, structure, and radiative properties. The Cloud-Aerosol Lidar with Orthogonal Polarization (CALIOP) is an elastically backscattered active polarization-sensitive lidar instrument onboard CALIPSO. CALIOP transmits laser light simultaneously at 532 and 1064 nm at a pulse repetition rate of 20.16 Hz. The lidar receiver subsystem measures backscatter intensity at 1064 nm and two orthogonally polarized components of 532 nm backscatter signal that provide the information on the vertical distribution of aerosols and clouds, cloud particle phase, and classification of aerosol size (Winker et al., 2010). The details of the data retrieval method are explained in Li et al. (2012).

## 2.4 The model simulations

The ECHAM6–HAMMOZ aerosol–chemistry-climate model used in the present study comprises of the ECHAM6 global climate model coupled to the two moment aerosol and cloud microphysics module HAM (Stier et al., 2005; Tegen et al., 2019) and the sub-model for trace gas chemistry MOZ (Kinnison et al., 2007). HAM predicts the nucleation, growth, evolution, and sinks of sulfate ($SO_4^{2-}$), black carbon (BC), particulate organic matter (POM), sea salt (SS), and mineral dust (DU) aerosols. The size distribution of the aerosol population is described by seven log-normal modes with prescribed variance as in the M7 aerosol module

(Stier et al., 2005; Zhang et al., 2012b). Moreover, HAM explicitly simulates the impact of
aerosol species on cloud droplet and ice crystal formation. Aerosol particles can act as cloud
condensation nuclei or ice nucleating particles. Other relevant cloud microphysical processes
such as evaporation of cloud droplets, sublimation of ice crystals, ice crystal sedimentation,
detrainment of ice crystals from convective cloud tops, etc. are simulated interactively
(Lohmann and Ferrachat, 2010; Neubauer et al., 2014). The anthropogenic and fire emissions
of sulfate, BC, and OC are based on the AEROCOM-ACCMIP-II emission inventory for the
study period 2010 – 2011 (Textor et al., 2006). The MOZ sub-model describes the trace gas
chemistry from the troposphere up to the lower thermosphere. The species included within the
chemical mechanism are contained in the $O_X$, $NO_X$, $HO_X$, $ClO_X$, and $BrO_X$ chemical families,
along with $CH_4$ and its degradation products. Several primary non-methane hydrocarbons
(NMHCs) and related oxygenated organic compounds are also included. This mechanism
contains 108 species, 71 photolytic processes, 218 gas-phase reactions, and 18 heterogeneous
reactions on aerosol (Kinnison et al., 2007). Details of anthropogenic, biomass burning,
biogenic, emissions fossil fuel sources, etc. are reported by Fadnavis et al. (2017a).

The model simulations are performed at the T63 spectral resolution corresponding to

1.875º × 1.875º in the horizontal dimension, while the vertical resolution is described by 47
hybrid σ-p levels from the surface up to 0.01 hPa. The model has 12 vertical levels in the
UTLS (50 – 300 hPa). The simulations have been carried out at a time step of 20 minutes.
AMIP sea surface temperature (SST) and sea ice cover (SIC) (Taylor et al., 2000) were used as
lower boundary conditions. We performed 10-member ensemble runs by varying the initial
conditions (both SST and SIC) starting between 1 and 10 January 2010 and ending on 31
December 2011 to obtain statistically significant results. The analysis is performed for the year
2011. The 2011 Indian monsoon was well within the long term norm, with no strong
influences from the Indian Ocean Dipole or El Niño modes of inter-annual climatic variability.
We refer to it as the control simulation (CTRL). In previous work, Fadnavis et al. (2013;
2017b) used the ensemble means from 6–10 members to analyze the variability of aerosols
and associated impacts during the monsoon season. In two emission sensitivity simulations we
have applied (1) a flat 48% increase in anthropogenic $SO_2$ emissions over India (referred to as
Ind48 simulation) and, (2) a flat 48% increase in anthropogenic $SO_2$ emissions over India and
a flat 70 % decrease in anthropogenic $SO_2$ emissions over China simultaneously, (referred to
as Ind48Chin70 simulation); same assumptions for simulated years. The simulation design is
based on the estimated trend of 4.8 % per year over India and -7.0 % over China, from OMI
$SO_2$ observations during 2007 – 2017. The Ind48 and Ind48Chin70 simulations are also 10
member ensemble runs for the same period as CTRL and are analyzed for the year 2011 (see
Table-1). We compare the CTRL and Ind48, Ind48Chin70 simulations to understand the
seasonal impact of enhanced sulfate aerosol on the UTLS, radiative balance, and cirrus clouds.
We should mention that our simulations are canonical in design in order to show the impact of
Asian sulfate aerosols; they do not include many of the observed complexities, like radiative
forcing due to non-sulfate aerosols (e.g., organics, nitrates, and dust, etc.). The QBO is not
internally generated in the model. Notwithstanding this, the present work provides valuable
insight into the relevance of the impact of sulfate aerosol originating from India and China on
the UTLS.
The seasons considered in this study are pre-monsoon (March-May), summer-
monsoon (June-September), post-monsoon (October-November), and winter (December-
February).

**2.5 Offline radiative calculations**

We use offline radiative calculations to explore the radiative impacts of enhanced sulfate aerosol loadings in the UTLS only (300 – 50 hPa), compared to the all atmosphere enhancement. Radiative effects associated with the sulfate aerosol enhancement are calculated using the SOCRATES radiative transfer model (Edwards and Slingo, 1996; Rap et al., 2013) with the CLASSIC aerosol scheme (Bellouin et al., 2011). We used the offline version of the model with six shortwave and nine longwave bands, and a delta-Eddington two-stream scattering solver at all wavelengths.

**3. Model evaluation with observations via remote sensing**

In Figs. 1a–h, we show the distribution of seasonal mean cloud ice mixing ratio from ECHAM6–HAMMOZ and combined measurements of total cloud ice from CloudSat and CALIPSO (2C–ICE) (2007 – 2010). Although cloud ice is underestimated in the model (~6–15 mg·kg$^{-1}$; 35–45%), the spatial distribution is well reproduced. Both the model simulations and the observations show high amounts of cloud ice in the mid-upper troposphere (450 – 250 hPa) over the Asian monsoon region (80 – 120 ⁰E). Cloud ice peaks during the monsoon season with a second peak in the pre-monsoon season. The observed seasonality might have linkages with seasonal transport process in the troposphere (details in section 4.2). The differences in model simulations and observations are due to uncertainties in satellite observations and model biases (Li et al., 2012); for example, the model does not consider large ice particles unlike the cloud ice measurement from CloudSat and CALIPSO. The total ice water mass estimates from 2C–ICE combine measurements from CALIPSO lidar

depolarization, which is sensitive to small ice particles (i.e., cloud ice represented in global
climate models), and CloudSat radar, which is very sensitive to larger ice particles (i.e.,
precipitating ice or snow) (Li et al., 2012).
Figures 2a-l shows the distribution of seasonal mean AOD from MISR (2000 – 2016),
model simulations (CTRL) and AERONET observations (2006 – 2016) (Bihar, Jaipur,
Kanpur, Karachi, XiangHe, NghiaDo). The model reproduces the large AOD over the Indo-
Gangetic Plains and Eastern China as seen in the MISR. However, simulated AOD is
underestimated in the model compared to MISR over the Indo-Gangetic Plains (~0.4) and
overestimated over Eastern China (~0.25). Comparison with AERONET observations also
shows underestimation in the model AOD over the stations in the Indo-Gangetic plains and
China (~0.23 – 0.35). The underestimation of model AOD over India and overestimation over
china in comparison with MISR is an agreement with ECHAM-HAMMOZ simulations in
Kokkola et al. (2018) and Tegen et al. (2019). The differences in the magnitude of AOD
between model, satellite remote sensing (MISR) and AERONET observations may be due to
various reasons, e.g., Satellite remote sensing detects AOD from top of the atmosphere while
AERONET detects AOD from the ground. Dumka et al. (2014) have documented that in
AERONET observations, the aerosols above 4 km contribute 50 % of AOD at Kanpur (in the
Indo-Gangetic plains). Inclusion of nitrate aerosol may affect the distribution of the AOD.
There are also uncertainties in model estimates of sea salt emission and parameterization
(Spada et al., 2013). The dust aerosols are underestimated the model (Kokkola et al., 2018).
The majority of CMIP5 models underestimate global mean dust optical depth (Pu and Ginoux,
2018). During the monsoon season, the large AOD values near 25 °N, 75 °E are likely due to
the presence of high amounts of sea salt and water-soluble aerosols in the model.

## 4. Results

### 4.1 A layer of aerosol in the UTLS

The Asian region (8 – 45 ºN; 70 – 130 ºE) experiences convective instability throughout the year with a peak in the monsoon season (Manohar et al., 1999; Luo, 2013). Distribution of seasonal mean outgoing longwave radiation, simulated ice crystal number concentration, and cloud droplet number concentrations representing convection is shown in Fig. S1. It depicts convection over the Asian region rising to the UT throughout the year and is wide-spread during the monsoon season. The summer-monsoon convection lifts the boundary layer aerosols to the upper troposphere, leading to the formation of the Asian Tropopause Aerosol Layer (ATAL) (Fadnavis et al., 2013, Vernier et al., 2015). The CALIPSO lidar and Stratospheric Aerosol and Gas Experiment II (SAGE-II) satellite observations reveal that the ATAL extends over a wider Asian region (15 – 40 °N, 60 – 120 °E) between 12 –18 km (Vernier et al., 2015; Fadnavis 2013).The ECHAM6-HAMMOZ simulations reproduce the formation of an ATAL (extinction and sulfate aerosol) in the UTLS during the summer-monsoon season (Figs. 3a-b). The aerosol layer in the UTLS is connected to the troposphere during the pre-monsoon, indicating transport of tropospheric aerosols into the UTLS. From March to November, the altitude of convective outflow propagates deeper into the UTLS. Strong uplift during the summer-monsoon season lifts the mid-tropospheric aerosols and aerosol precursors to the UTLS, generating aerosol minima in the mid-troposphere (Fadnavis et al., 2013). During the summer-monsoon season, the convective transport mostly occurs from the Bay of Bengal, the South China Sea and southern slopes of Himalayas (Fadnavis et al., 2013; Medina et al., 2010). After the convective uplift, at altitudes above ~360 K, radiatively driven upward transport in the anticyclonic monsoon circulation occurs at a rate of

~1 K·day$^{-1}$; this is a slower uplift than convection but faster than outside the anticyclone
(Vogel et al., 2019). The simulated distribution of aerosol extinction and sulfate aerosols at
100 hPa from the CTRL simulation shown in Figs. 3c-d indicates maxima in aerosol extinction
(Fig. 2c) and sulfate aerosols (Fig. 2d) in the anticyclone region.
The estimated ratio of ECHAM6–HAMMOZ simulated sulfate aerosols in the UTLS to
the total aerosol amount is 6:10 pointing at sulfate aerosols as a major ATAL constituent.
Balloonsonde observations over South Asia also indicate that large amounts of sulfate aerosols
may be present in the ATAL (Vernier et al., 2015). Tropospheric $SO_2$ and sulfate aerosol
transported into the stratosphere during volcanically quiescent periods are potentially large
contributors to the stratospheric aerosol burden (SPARC-ASAP, 2006).
**4.2 Transport into the upper troposphere and lower stratosphere**
We investigate the transport pathways of sulfate aerosol during different seasons from
anomalies of sulfate aerosol for (1) Ind48, and (2) Ind48Chin70 simulations. Firstly, we
present a vertical distribution of anomalies (relative to CTRL) of sulfate aerosol for Ind48
simulations in Figs. 4 a-h. The striking feature is poleward transport of Indian emissions in the
UTLS throughout the year. A layer of sulfate aerosols enhancement extending from India to
the Arctic (68 – 90 °N), is seen near the tropopause, during pre-monsoon (3 – 15 ng·m$^{-3}$) and
the lowermost stratosphere during summer-monsoon (2 – 15 ng·m$^{-3}$), post-monsoon (2 – 6
ng·m$^{-3}$) and winter (0.5 – 3 ng·m$^{-3}$) seasons. This layer may be due to transport of Indian
sulfate aerosols to the Arctic by the lower branch of the Brewer-Dobson circulation. These
sulfate aerosols enhance the AOD in the UTLS by 0.184E-04 (i.e. 1.1%) to 4.15E-04 (i.e.
4.17%) over India and the Arctic (seasonal details in Table-2). Past studies also indicate the
transport of pollution from South Asia and East Asia to the Arctic predominantly in the UTLS
(Shindell et al., 2008; Fisher et al., 2011). From multi-model simulations, Shindell et al.
(2008) show that seasonally varying transport of south-Asian sulfate aerosols to the Arctic
maximizes in the pre-monsoon season. This enhancement of sulfate aerosols that maximizes
during the pre-monsoon is also illustrated in Figure 4a.

Figure 4 also shows that during most seasons the vertical transport occurs from the Bay

of Bengal, Arabian Sea, southern slopes of Himalayas (60 – 100 °E; 15 – 35 °N), except
during the post-monsoon season when it occurs from the west Asia and Tibetan Plateau region
(20 – 35 °N; 60 – 95 °E). This may be due to the transport of sulfate aerosols from India to
these regions, which might have been lifted to the UTLS by the post-monsoon convection (see
Figs. S1 c, h, k, and S2 c). The enhancement of sulfate aerosols in the monsoon anticyclone
(an ATAL feature) and the cross-tropopause transport associated with the summer monsoon
convection is evident in Figs. 4c-d (enhancement ~5 – 15 ng·m$^{-3}$; 10 – 36 %). Past studies
show that the aerosols transported into the lower stratosphere by the monsoon convection are
recirculated in the stratosphere by the lower branch of the Brewer-Dobson circulation (Randel
and Jensen, 2013; Fadnavis et al., 2013; Fadnavis et al., 2017b). Yu et al., (2017) report that
~15 % of the Northern Hemisphere column stratospheric aerosol originates from the Asian
summer monsoon anticyclone region. Figure 4d shows that aerosols spread to east and west
from the anticyclone (20 – 120 °E), likely due to east/westward eddy shedding from the
anticyclone (Fadnavis and Chattopadhyay, 2017; Fadnavis et al., 2018). Eddy shedding is not
evident in the seasonal mean distribution (Fig. 3 b) due to its short duration (i.e., days) and
episodic nature.
The influence of the Chinese $SO_2$ emission reduction (Ind48Chin70) on the vertical
distribution of sulfate aerosols is shown in Figs 5a-h. In the pre-monsoon season, the transport
pattern is similar to the Ind48 simulations; however, the enhancement of sulfate aerosols at the
Arctic tropopause is significantly hindered (1 – 3 ng.m$^{-3}$). The subsidence over north India (20
– 35 °N) has resisted sulfate aerosols crossing tropopause (Figs. 9 a, e). A feeble plume tilted
westward is seen during the monsoon season (Figs. 5c-d) and eastward-equatorward during
post-monsoon due to changes in circulations (ascending winds over south India and strong
subsidence over north India; Figs. 9 f-g). Entrainment into the anticyclone and cross-
tropopause transport of the sulfate aerosols, seen in the Ind48 simulation, is inhibited by this
subsidence. Interestingly, during summer-monsoon and post-monsoon seasons, poleward
transport of south Asian sulfate aerosols have also been cut-off due to circulation changes
(subsidence over north India see below in Figs. 9f-g). During winter, vertical winds over ~20
°N lifts aerosols from India to the mid-troposphere and further transported to the Arctic (Figs.
5 k-l, Fig. 9h). The vertical transport of sulfate aerosols increases AOD in the UTLS over
India by ~0.32E-04 (0.61 %) to 19.20E-04 (19.25 %) (except winter) and Arctic by 2.09E-04
(16.45 %) during the pre-monsoon season (see Table-2).

**5. Impact of changes in $SO_2$ emissions**
**5.1    Radiative forcing**
The seasonal mean anomalies of net radiative forcing at TOA due to sulfate aerosols
from the Ind48 and Ind48Chin70 simulations of the ECHAM6-HAMMOZ model are
illustrated in Figs. 6a-h. In general, both simulations show negative forcing over India and the
surrounding region where sulfate aerosols are dispersed during that season (-0.2 to -2 $W{\cdot}m^{-2}$).
Distribution of anomalies of sulfate aerosols at 850 hPa (Figs. S2 a-d) and Figs. 4 a-d show
that in the Ind48 simulations, during all seasons, sulfate aerosols are transported south-west
over the Arabian Sea and partially to the east (during pre-monsoon, monsoon, and winter
towards Myanmar; during post-monsoon and winter to North-east China). These regions are
associated with negative radiative forcing for Ind48 in Figs. 6 a-d. This negative radiative
forcing extending from North India towards the Arctic during pre-monsoon and summer-
monsoon is likely due to the poleward transport of south Asian sulfate aerosols in the UTLS (2
– 10 $\mu g.m^{-3}$) reflecting back solar radiation (see Figs. 4a, c). The poleward extension of
negative RF is not evident during the post-monsoon and winter seasons (Figs. 6 c, d). This
may be due to fine and thinner sulfate aerosol layer (~1 – 4 $\mu g.m^{-3}$) in the upper troposphere
which partially reflect back solar radiation, leading to weak positive and negative RF (-0.1 to
+0.5 $W{\cdot}m^{-2}$) over mid-high latitudes (40 – 70ºN).

The simulated RF at TOA in the Ind48Chin70 simulations is negative over India

during all seasons (~-0.6 to -2 $W{\cdot}m^{-2}$) (Figs. 6e-h) similar to Ind48 (Figs. 6a-d). In addition,
the Chinese $SO_2$ emission reductions in Ind48Chin70 have produced a significant positive
forcing ~0.6 to 6 $W{\cdot}m^{-2}$ over China (100 – 140 ºE). The positive RF is also seen over the
western Pacific (pre-monsoon, summer-monsoon, and winter) and Bay of Bengal (post-
monsoon and winter). This is due to the negative anomalies of sulfate aerosols over these
regions in Ind48Chin70 (Figs. S2 e-h). The south-west ward transport of Indian sulfate
aerosols to the Arabian Sea in the lower troposphere (Figs. S2 e-h) during all seasons
producing a negative RF in that region is evident in Figs. 6.e-h. During the monsoon season,
the narrow localized plume leads to a negative regional forcing (30 – 40 ºN, 80 – 95 ºE) of ~-
0.6 W.m$^{-2}$. The negative RF near 40 – 50 ºN may be due to sulfate aerosols in the lower
troposphere (Fig. 5c). The negative RF values (-0.1 to -0.4 W·m$^{-2}$) extending from the Indian
region to the Arctic are likely due to the poleward transport in the upper troposphere during
the pre-monsoon season and in the lower-mid troposphere during the winter season (Figs. 6 e,
h). The seasonal mean net radiative forcing due to sulfate aerosols at the surface and at TOA
are similar for both the Ind48 and Ind48Chin70 simulations (Figs. S3 a-h), due to the strong
scattering properties of the sulfate aerosols (Forster et al., 2007).

The comparison of RF at the TOA obtained from ECHAM6–HAMMOZ simulations

over the Arabian Sea (60 – 75 ºE, 0 – 20 ºN) during winter (Ind48: -2.0 W·m$^{-2}$, Ind48Chin70:
1.5 W·m$^{-2}$) (Fig. 4a) show reasonable agreement with the INDOEX experiment (-1.25 to -2.0
W·m$^{-2}$ over North India during January – March 1999 (Verma et al., 2012). Yu et al. (2016)
reported that the increase in sulfate AOD (0.06 – 0.15) over the tropics (30 °S – 30 °N) since
the pre-industrial period has exerted a forcing of -0.6 to -1.3 W·m$^{-2}$.

The corresponding distribution of sulfate aerosol DRF at TOA estimated with our

offline simulations for the four seasons for Ind48 and Ind48Chin70 are shown in Figs. 6 i-p.
The results from the offline model are in reasonable agreement with the ECHAM6-HAMMOZ
simulations, although their magnitude differs spatially. Both the Ind48 and Ind48Chin70
simulations have produced negative RFs, varying between -0.2 and -2.0 W·m$^{-2}$ over India. The
reduction of SO$_2$ emission over China leads to an increase in RF of 2 – 6 W·m$^{-2}$, comparable
with the corresponding values simulated in ECHAM6–HAMMOZ. The differences in
estimated RF in the offline calculations and the ECHAM6–HAMMOZ simulations are likely
due to the fact that the implicit dynamical responses in ECHAM6–HAMMOZ are not captured
in the offline simulations. However, the offline calculations are important insofar as they
isolate the direct radiative impact of the simulated changes in aerosol loading.

The offline calculations further allow the specific effect of the enhanced aerosol layer

in the UTLS (300-50 hPa) to be discriminated (Figs. 7a-h). Figures 7a-d shows the direct
radiative forcing at TOA (estimated from our offline simulations) induced by the sulfate
aerosol enhancement in the UTLS (300 – 50 hPa) during the four seasons. The RF values from
Ind48 are mostly negative over India, China and extending to the Arctic (~-0.001 to -0.015
$W \cdot m^{-2}$), due to the presence of the sulfate aerosol plume in the UTLS. Interestingly, the
Ind48Chin70 simulation also shows negative RFs in the region co-located with the UTLS
plume, e.g. in the summer-monsoon season, the plume over north India leads to negative RF
values. Similarly, in the post-monsoon season, the sulfate aerosols plume extends to 15S and
leads to negative RF values (~ -0.001 to -0.005 $W \cdot m^{-2}$) (see Fig 7g and Fig. S4). In the pre-
monsoon season, the aerosol plume travels to the Arctic below or near the tropopause,
therefore partial contribution to RF from the UTLS (300 to 50 hPa) might have produced
positive anomalies of 0.0001 to 0.0005 $W \cdot m^{-2}$ in mid-high latitudes. During winter, sulfate
aerosols do not reach above the tropopause (Figs. 5 g-h) and therefore RF values are positive
over India and China. Thus the radiative forcing caused specifically by UTLS aerosol shows a
much clearer signal than the forcing due to the entire aerosol column (compare Figs. 6 and 7a-
h). The sulfate aerosol layer, corresponding to the ATAL in the summer monsoon season, in
the Ind48 simulation leads to a RF of ~-0.011 to -0.015 $W \cdot m^{-2}$ (Fig.7b). It is reduced to -0.001
to -0.003 $W \cdot m^{-2}$ in the Ind48Chin70 simulations (Fig.7f) due to reduction of transport of
sulfate aerosols in the UTLS. The short term ATAL RF at TOA has previously been estimated
as about ~-0.1 $W \cdot m^{-2}$ over the Asian region during 1998 – 2015 (Vernier et al., 2015). The
radiative forcing reported here caused solely by the sulfate aerosol particles in the UTLS is
lower than the value reported by Vernier et al. (2015), who give an integral value for the
ATAL and not only for the sulfate particles.

**5.2 Incoming solar radiation, temperature, and stability of the troposphere**

An important impact of sulfate aerosols in the atmosphere is solar dimming, which
counteracts the surface temperature response to the anthropogenic $CO_2$ increase (Ramanathan
et al., 2005). There is observational evidence (1300 sites globally) indicating that one-third of
potential continental warming attributable to increased greenhouse gas concentrations has
been compensated by aerosol cooling during 1964 – 2010 (Storelvmo et al., 2016). Solar
radiation measurements over the Indian region (at 12 stations) during 1981 – 2004 show a
declining trend varying between -0.17 to -1.44 $W \cdot m^{-2} yr^{-1}$ (Padma Kumari et al., 2007). While
not directly comparable to these previous studies, Ramanathan et al. (2005) reported a
negative trend in solar flux observations at 10 different Indian stations (-0.42 $W \cdot m^{-2}$) and their
model simulations show a trend of -0.37 $W \cdot m^{-2}$ induced by the changes inBC and sulfate
aerosols over India (0 – 30 °N and 60 – 100 °E).
We estimate the changes in net solar radiation at the surface for four seasons from the
Ind48 and Ind48Chin70 simulations. Figures 7i-l shows that the Ind48 simulations have
produced negative anomalies in net solar radiation (SR) at the surface (~-0.5 to -3 $W \cdot m^{-2}$) over
India and parts of China (where sulfate aerosols are transported) due to the enhanced sulfate
aerosol layer reflecting back solar radiation. In general, the seasonal mean distribution of
anomalies in net solar radiation at the surface is similar to the distribution of the anomalies in
RF at the TOA. Reduction of Chinese $SO_2$ emissions along with an increase of $SO_2$ emissions
over India (Ind48Chin70) has produced a reduction of solar radiation over India while there is
a significant increase over China ($1 - 5$ W·m$^{-2}$) (see Figs. 7 m-p).
Sulfate aerosols also absorb infrared radiation thus causing heating locally and
producing a cooling in the region below by solar dimming (Niemeier and Schmidt, 2017).
Therefore, seasonally varying transport of sulfate aerosol may affect the thermal structure in
the receptor region. Figure 8 shows a temperature enhancement near the region of transport of
sulfate aerosols in the UTLS and a cooling of the atmosphere below it. For example, in the
Ind48 simulations, positive temperature anomalies are seen near the sulfate aerosol layer
extending to the Arctic, with negative anomalies below the layer during all seasons (except
winter) (Figs. 8 a-h). Similarly, a warming ~$0.1 - 0.7$ K over India simulated in the
Ind48Chin70 simulations in pre-monsoon and post-monsoon (Figs. 8 i-j, m-n). During winter,
in the Ind48Chin70 simulation, poleward transport occurs from the Indian lower/mid-
troposphere to the lower stratosphere of mid-high latitudes. This region shows positive
anomalies of temperature ~0.2 to 1K (see Figs. 8 o-p and Figs. 5 g-h).
As shown in Figure 8 the amplitude of the temperature anomalies in the UTLS varies
seasonally and regionally. In general, there is temperature enhancement in the UTLS over
North India and South China ($20 - 35$ °N, $75 - 130$ °E) of  ~$0.2\pm0.15$ to $0.8\pm0.72$ K in Ind48
(all four season) and ~$0.1\pm0.08$ to $0.5\pm0.23$ K in Ind48Chin70 (pre-monsoon and post-
monsoon). Temperature uncertainties in this paragraph are obtained by determining the
variability within the 10-member ensemble. After reaching the Arctic, these sulfate aerosols
cause substantial warming in the lower stratosphere i.e. ~$1\pm0.62$ to $1.6\pm1.07$ K in Ind48 during
all seasons and $0.7\pm0.60$ to $1.6\pm1.43$ K in Ind48Chin70 in pre-monsoon and winter seasons.
Figure 8 also shows reduction in temperature of -0.1±0.05 to -0.6 ± 0.4 K in the troposphere,
below the warming, corresponding to the UTLS sulfate aerosols layer.

The changes in the circulation are illustrated in Figs. 9a-h. It shows ascending winds in

the region of the sulfate aerosol plume. For example the Ind48 simulations show ascending
winds over northern India (while there is subsidence in the upper troposphere over 10 – 30 ºN)
during all seasons and in the Ind48Chin70 simulations during the pre-monsoon season. The
reduction of Chinese $SO_2$ emissions (Ind48Chin70) induces strong descending winds over
northern India during the summer-monsoon and post-monsoon. It hindered the poleward
transport of the plume as discussed in section 4.2.

The sulfate aerosol-induced cooling in the upper troposphere (below the layer of

sulfate aerosols) and subsidence in the upper troposphere cause a stabilization of the upper
troposphere (Pitari et al., 2016). Figures 9 i-p shows that anomalies of Brunt-Väisälä
frequency are positive $(0.2 – 3 \text{ s}^{-1} \times 10^{-5})$ in the upper troposphere (250 – 150 hPa) over north
India and south China (20 – 35 °N, 70 – 130 °E) during all the seasons in Ind48 and for the
pre-monsoon and post-monsoon season in the Ind48Chin70 simulations. Thus enhanced Indian
sulfate aerosols have increased the stability of the upper troposphere and produce a cooling of
~0.2 – 1.2K (Fig.8) in the upper troposphere. They have induced upper tropospheric
subsidence (10 – 30 °N) in Ind48 and ind48Chin70 simulations (except in winter in
Ind48Chin70). Upper tropospheric temperature and stability play important roles in rainfall
suppression (Wu and Zhang, 1998; Fadnavis and Chattopadhyay, 2017). Thus upper
tropospheric cooling and enhanced stability may suppress the rainfall over India in all seasons
in Ind48 and in the pre-monsoon and post-monsoon season in the Ind48Chin70 simulations.
However, a complete analysis of the impact of the enhanced surface aerosols on rainfall is
beyond the scope of this study.

## 5.3 Cirrus Clouds

Cirrus clouds cover at least about 30 % of the Earth's area on annual average (Stubenrauch
et al., 2013, Gasparini et al., 2018), occurring mainly between 400 – 100 hPa altitude. They play
an important role in the Earth's energy budget (Gasparini and Lohmann, 2016; Hartmann et al.,
2018), in transport of water vapor into the stratosphere (Randel and Jensen, 2013), as well as in
the atmospheric heat and energy cycle (Crueger and Stevens, 2015). Cirrus clouds can form by
either homogeneous nucleation by freezing of dilute sulfate aerosols or by heterogeneous ice
nucleation in the presence of ice nuclei, most commonly dust (Ickes et al., 2015; Cziczo et al.,
2017). Moreover, a large fraction of cirrus clouds have a liquid origin as the ice crystals were
either nucleated at mixed-phase conditions and transported to lower temperatures or detrained
from convective cloud tops (Krämer et al., 2016; Wernli et al., 2016; Gasparini et al., 2018). All
mentioned formation processes except heterogeneous nucleation of ice crystals below the
homogeneous freezing temperature (i.e. at cirrus conditions) are represented in by our model
simulations. However, heterogeneous freezing on dust and black carbon aerosols is included in
mixed-phase clouds (Lohmann and Hoose, 2009), for temperatures between freezing and -35°C.
Figures 10 a-h shows the impact of $SO_2$ emission changes on cirrus clouds. It shows a decrease
(5 – 30 %) of cirrus clouds over North India (20 – 35 °N) in the UTLS. The decrease in cirrus
clouds coincides with a significant decrease of ice crystal number concentration by -0.15 to -0.5
$cm^{-3}$ between 250 – 50 hPa (except in winter in Ind48Chin70 since the plume of sulfate aerosols
does not reach the upper troposphere) (Figs. 10i–p).
Our analysis indicates that an increase in the upper tropospheric sulfate aerosol
concentration leads to a temperature increase in the upper troposphere and lower stratosphere
of about ~0.2±0.15 to 0.8±0.72 K over north India and South China and to a cooling below
(Fig. 8). This temperature changes causes a decrease in the upper tropospheric temperature
gradient and vertical velocity, concurrently an increase in the upper tropospheric (200 – 100
hPa) static stability (Brunt–Väisälä frequency) (over 80 – 120 °E) (Figs. 9 i-p) (Figs. 9 a-h). A
combination of decreased upper tropospheric updraft motion and increased temperature
decreases the likelihood of cirrus cloud formation in a similar way as for the simulated
responses to volcanic eruptions or stratospheric sulfur geoengineering (Kuebbeler et al., 2012,
Pitari et al. 2016, Visioni et al., 2018a).

**6. Discussion**

Our model simulations presented here provide seasonal transport processes and
estimates of radiative forcing for the year 2011. The inter-annual variability in the transport
processes may impact the injection of sulfate aerosols shallow/deep into the lower
stratosphere. The stratospheric warming produced in response to the transport of rising South
Asian anthropogenic sulfate aerosol in the UTLS over Asia and further to the Arctic (Fig. 4
and Fig.5) may modulate the Quasi-biennial Oscillation (QBO) and thereby the transport of
sulfate aerosol from the tropics to the extra-tropics. The QBO phases are modulated by the
amount of sulfate and height of the injection (Aquila et al., 2014; Niemeier and Schmidt,
2017; Visioni et al., 2018b). A previous study reports that the QBO slows down after an
injection of 4 Tg (S) yr$^{-1}$ into the stratosphere and completely shuts down after the injection of
8 Tg (S) yr$^{-1}$ (Niemeier and Schmidt, 2017). However, another model study finds that the
QBO, even for a larger amount of SO$_2$ injections, does not deviate much from present day
conditions (Richter et al., 2018). These studies indicate that there is a complicated interaction
between UTLS aerosols, atmospheric dynamics and atmospheric chemistry (Richter et al.,
2017; Niemeier and Schmidt, 2017; Visioni et al., 2018b). The QBO is known to modulate the
tropical convection (Collimore et al., 2003; Fadnavis et al., 2013; Nie and Sobel, 2015). Thus
transport of sulfate aerosols into the stratosphere would impact the tropospheric hydrological
cycle in addition to the tropospheric aerosol loading. The increasing amounts of tropospheric
sulfate aerosol loading are linked with droughts via changes in radiative forcing, stability, and
tropospheric temperature gradient (Yeh et al., 2015; Kim et al., 2016). Simulations for a longer
time period and with the inclusion of QBO phases may reveal the influence of current SO$_2$
emission on tropospheric-stratospheric dynamics and the hydrological cycle. Nonetheless, the
results of the current study show the impacts of sulfate aerosols on the UTLS for realistic
emission perturbations over India and China.

**7. Conclusions**
This study investigated the long range transport of Asian sulfate aerosols and their
associated impacts on radiative forcing, temperature, circulation and cirrus clouds using
ECHAM6–HAMMOZ model simulations. We considered emissions perturbations of
anthropogenic SO$_2$ derived from OMI observations, namely (1) enhancement over India by 48
% (Ind48) and (2) enhancement over India by 48% and reduction over China by 70 %
simultaneously (Ind48Chin70). The Ind48 simulations show long-range transport of sulfate
aerosols from the Indian boundary layer (75 – 95 °E, 20 – 35 °N) to the UTLS and further
horizontally to the Arctic throughout the year. The reduction of Chinese $SO_2$ emissions inhibits
the transport of sulfate aerosols from India to the Arctic in the summer-monsoon and post-
monsoon seasons via subsidence over north India, which is induced in response to emission
perturbation. The enhancement of Indian emission increases the aerosol burden (AOD) in the
UTLS over North India by 0.184E-04 (1.1 %) to 19.20E-04 (19.25 %) and Arctic by 0.17E-04
(3.3 %) to 2.09E-04 (16.45 %). This leads to a warming (~0.2±0.15 to 0.8±0.72 K) in the
UTLS near the sulfate aerosol layer and to a cooling below it in the troposphere (0.1±0.05 to -
0.6 ± 0.4 K). It produces a negative net radiative forcing at TOA -0.2 to -2 $W{\cdot}m^{-2}$ over North
India. There is a substantial increase of ~ 0.6 to 6 $W{\cdot}m^{-2}$ in net radiative forcing at TOA over
China in response to the reduction of Chinese $SO_2$ emissions.

The RF at the TOA estimated from the offline radiative transfer model for

enhancement of Indian $SO_2$ emission is -0.2 to -2.0 $W{\cdot}m^{-2}$ over India. The reduction of $SO_2$
emissions over China leads to an RF of 2 to 6 $W{\cdot}m^{-2}$. These values are comparable with
results of the ECHAM6–HAMMOZ simulations, with the minor differences likely due to the
implicit dynamical impacts in response to enhanced south Asian $SO_2$ emissions in ECHAM6–
HAMMOZ not being represented in the offline model. The enhancement of sulfate aerosols in
the UTLS (300 – 50 hPa) produces a negative forcing in the region co-located with the aerosol
sulfate layer in the UTLS, extending from India to the Arctic in the Ind48 (-0.003 to -0.015
$W{\cdot}m^{-2}$) and the Ind48Chin70 (-0.001 – -0.005 $W{\cdot}m^{-2}$) simulations. The ATAL (due to sulfate
aerosols only) in the Ind48 simulation has produced an RF over north India of ~-0.011 – 0.015
$W{\cdot}m^{-2}$ (Fig.7b), which has reduced to -0.001 – -0.003 $W{\cdot}m^{-2}$ in the Ind48Chin70 simulation
(Fig.7f). This reduction is attributed to the subsidence over north India produced by the
Chinese $SO_2$ emission reduction.
An enhancement of 48 % in South Asian anthropogenic sulfate aerosols leads to a decrease in
cirrus clouds, cooling of the mid-upper troposphere over the northern regions of India and
south China throughout the year. This enhances the stability (anomalies in Brunt Väisälä
frequency 0.2 to 2 $s^{-1} \times 10^{-5}$) of the upper troposphere (~ 250  hPa) of these regions. Reduction
of Chinese $SO_2$ emissions does not stabilize the upper troposphere during the monsoon and
winter seasons since subsidence over North India inhibited the vertical transport of sulfate
aerosols to the UTLS. Upper tropospheric temperature and stability play an important role in
rainfall reduction. Strong subsidence, mid-upper tropospheric cooling and enhanced stability
over India may cause rainfall deficit (Wu and Zhang, 1998; Fadnavis et al., 2017c). The link
between these features and Indian rainfall deficit should be addressed in future research. It is
important to note that an increase in surface emissions of $SO_2$ does not necessarily lead to a
reduction in RF (as might be expected) but that regional enhancements of RF might occur in
response to an inherent dynamical response (including changes in high cloud cover) to
enhanced $SO_2$ emissions.

Data      availability:      OMI      $SO_2$      data     can     be     obtained     from
https://disc.gsfc.nasa.gov/datasets/OMSO2e_V003/summary?keywords=aura, MISR data is
available at https://giovanni.gsfc.nasa.gov/giovanni/, CALIPSO, and CloudSat measurements
can be obtained from http://www.cloudsat.cira.colostate.edu/data-products/. These satellite
data sets are freely available.
Author contributions: S.F. designed the study and wrote the paper, G.K. analyzed the model
simulations, M.R and A.R. performed offline radiative forcing computations. J.-Li provided
CALIPSO data. B.G and A.L. helped with aerosols and cirrus cloud analysis. R.M. contributed
to the analysis of the model results and the writing of the manuscript.
Competing interests. The authors declare that they have no conflict of interest.
*Acknowledgments*: Suvarna Fadnavis acknowledges Prof. Ravi Nanjundiah, Director of IITM,
with gratitude for his encouragement during this study. The authors thank the anonymous
reviewers for valuable suggestions and the high-performance computing team at IITM for
supporting the model simulations.

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

Table 1: Details of model simulations performed.

| Sr. No | Experiment description | Name of experiment | SST and Sea Ice | Initial condition of the simulation | Analysis is performed for period |
|---|---|---|---|---|---|
| 1. | Control simulation | CTRL | Monthly varying SST and Sea ice | 1 – 10 January 2010 | January – December 2011 |
| 2. | The anthropogenic emissions of $SO_2$ over India (8 – 40°N; 70 – 95°E) are increased by 48%. | Ind48 | Monthly varying SST and Sea ice | 1 – 10 January 2010 | January – December 2011 |
| 3 | The anthropogenic emissions of $SO_2$ over India (8 – 40°N; 70 – 95 °E) are increased by 48 % and reduced over China (23 – 45 °N; 95 – 130 °E) by 70 %. | Ind48Chin70 | Monthly varying SST and Sea ice | 1 – 10 January 2010 | January – December 2011 |



Table 2: Seasonal mean AOD in the UTLS (300 – 90 hPa) over India (75 – 95 ºE; 20 – 35 ºN)
and Arctic (75 – 97 ºE; 65 – 85 ºN) from simulations performed. AOD is calculated at different
altitude ranges indicated in brackets for some seasons since sulfate aerosol layer vary in altitude
in the UTLS.

| Season | AOD in the UTLS over India from Ind48 (AOD*1E-04) | AOD in the UTLS over India from Ind48Chin70 (AOD*1E-04) | AOD in the UTLS over Arctic from Ind48 (AOD*1E-04) | AOD in the UTLS over Arctic from Ind48chin70 (AOD*1E-04) |
|---|---|---|---|---|
| Pre-monsoon | 4.15 (4.17 %) | 19.20 (19.25 %) | 0.208 (0.017 %) (300–150 hPa) | 2.09 (16.45 %) |
| Summer-monsoon | 1.035 (2.17 %) | 6.14 (12.9 %) | 2.09 (2.14%) | -0.71 (0.073 %) |
| Post-monsoon | 0.462 (3.03 %) | 0.32 (0.61 %) | 0.17(3.3 %) (100–50 hPa) | -0.4.9 (-5.8 %) |
| Winter | 0.184 (1.1 %) | -1.01 (-6.62 %) | 1.47 (4.8 %) | -2.3 (-7.79 %) |










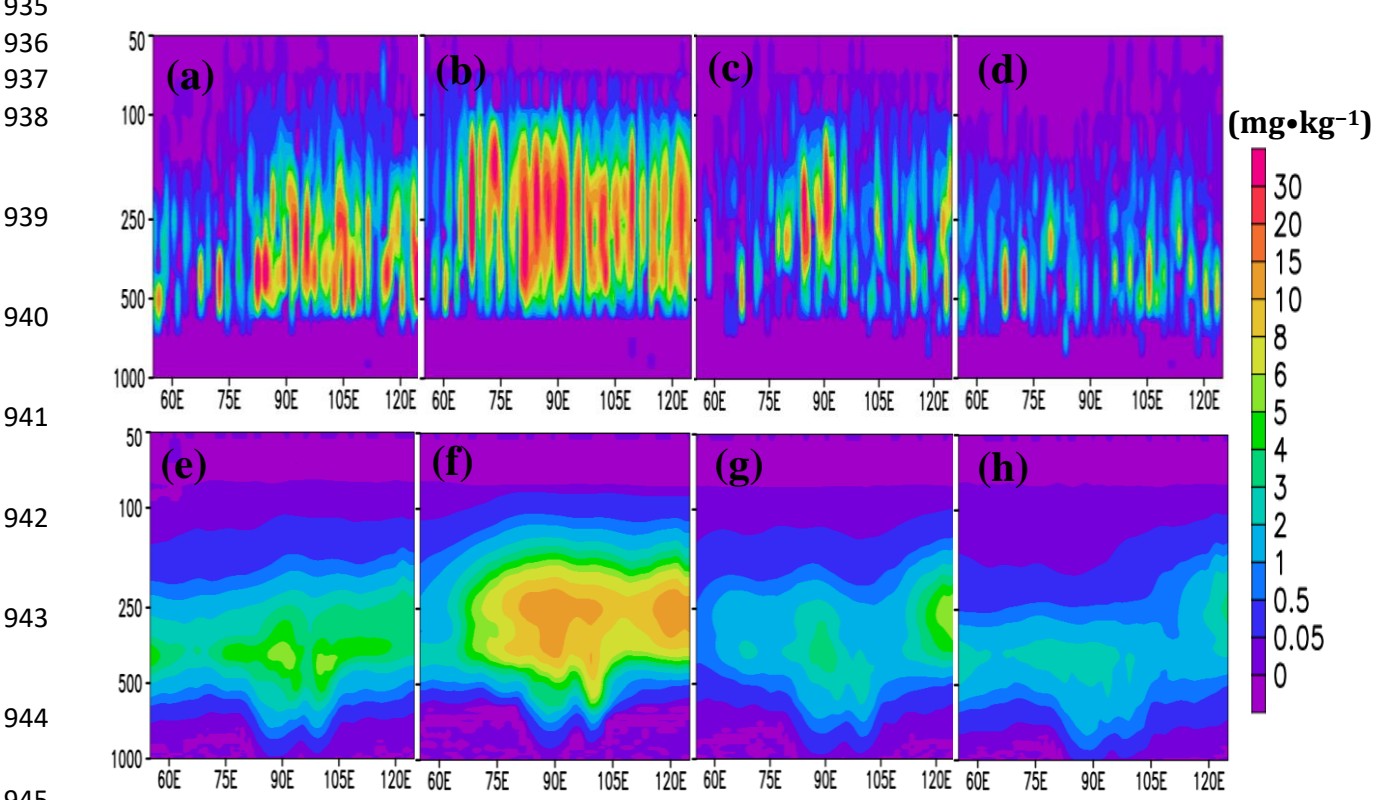

Figure 1: Seasonal mean distribution (2007 – 2010) of cloud ice mass mixing ratio (mg·kg$^{-1}$) from CloudSat and CALIPSO combined 2C–ICE L3 averaged for 20 – 40 °N for the (a) pre-monsoon, (b) summer-monsoon, (c) post-monsoon, and (d) winter season, (e)-(h) same as (a)-(d) but from CTRL simulations.





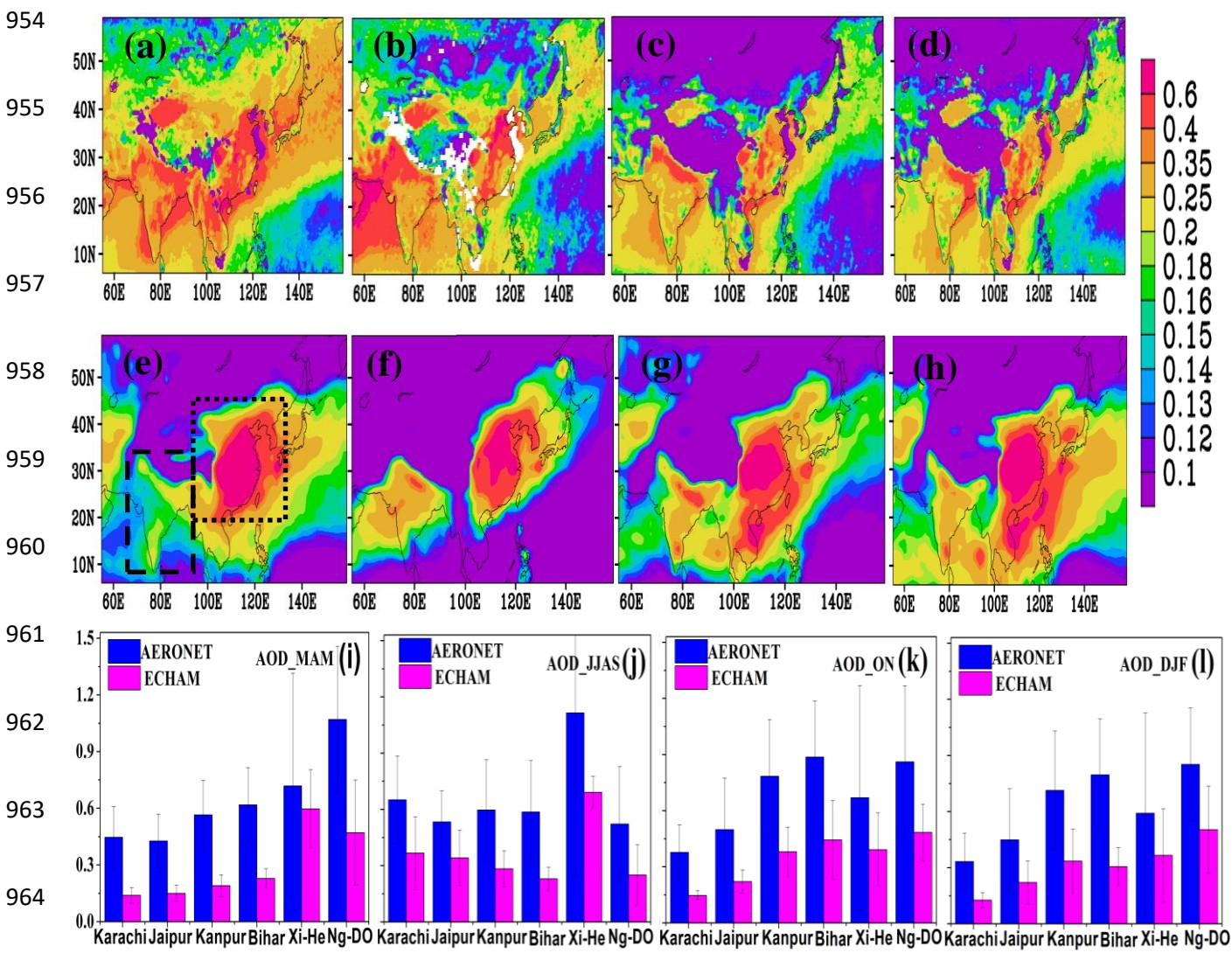

Figure 2: Seasonal mean Aerosol Optical Depth (AOD) from MISR (2000 – 2016) for the (a) pre-monsoon, (b) summer-monsoon, (c) post-monsoon, and (d) winter season, (e)-(h) same as (a)-(d) but from CTRL simulations, (i)-(l) same as  (a)-(d) but from AERONET (2006 – 2016) at the stations: Karachi, Jaipur, Kanpur, Bihar, Xiang-He, Nghia-Do. The dashed box in Fig. (e) indicates the South Asian region (70 – 95 °E, 8 – 35 °N) where $SO_2$ emissions are enhanced by 48 % and the dotted box indicates Chinese region where $SO_2$ emissions are reduced  by 70 % (95 – 130 °E; 20 – 45 °N).












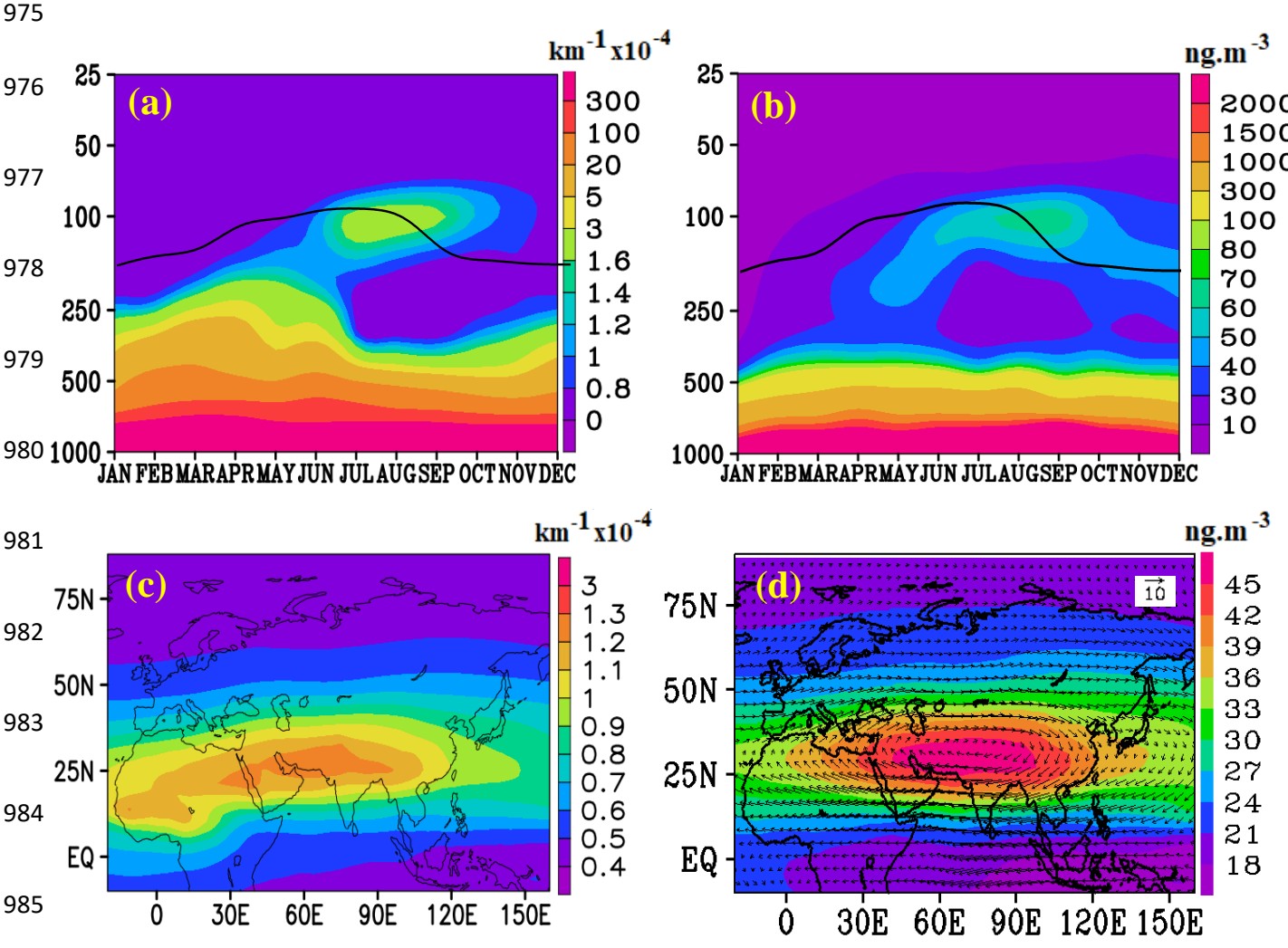

Figure 3: Monthly vertical variation of (a) extinction () averaged for 70 – 120 °E, 25 – 40 °N,
(b) same as (a) but for sulfate aerosols (ng·m$^{-3}$), (c) distribution aerosol extinction (km$^{-1}$× 10$^{-4}$)
at 100 hPa averaged for the summer-monsoon season, (d) distribution of sulfate aerosol (ng·m$^{-}$
$^{3}$) at 100 hPa averaged for the summer-monsoon season. Wind vectors in Fig. (d) indicate
extent of the anticyclone. Figs. (a)–(d) are obtained from CTRL simulations. Black line in (a)
and (b) indicates the tropopause.




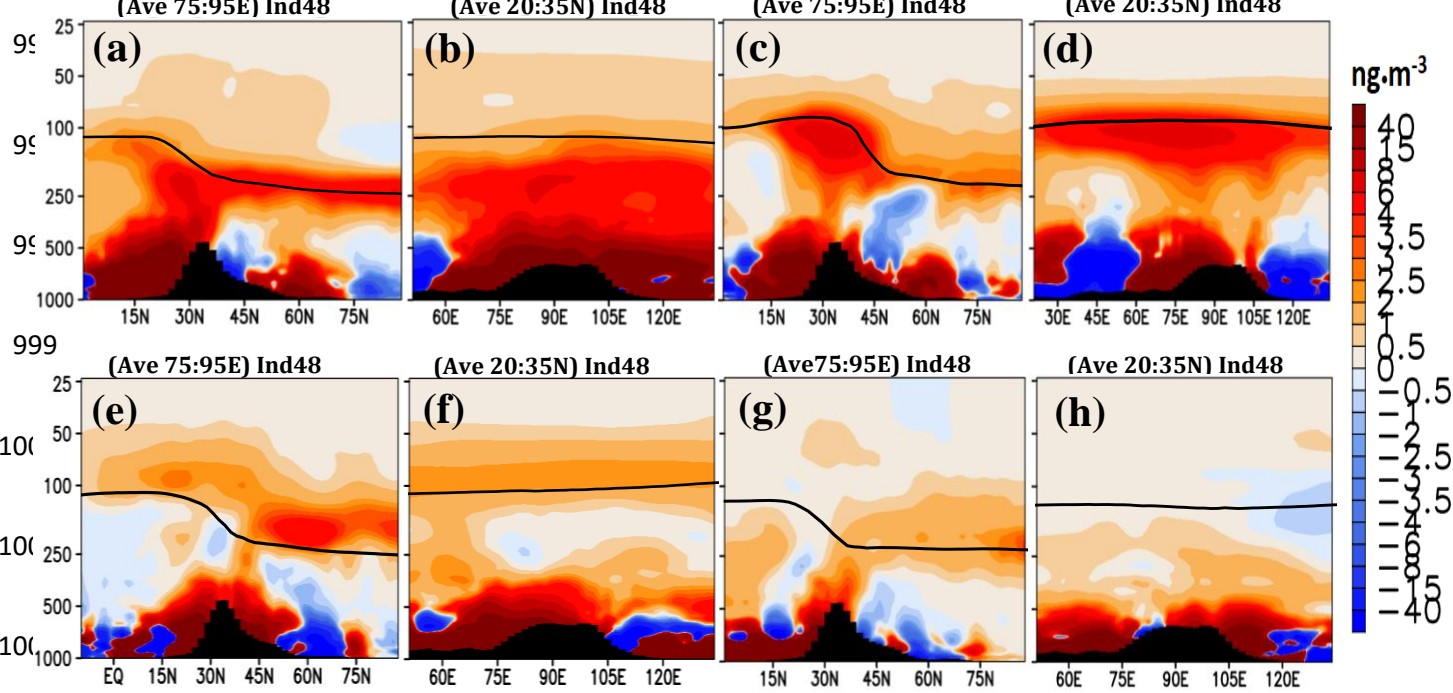

Figure 4: Vertical cross-section of anomalies in sulfate aerosols (ng·m$^{-3}$) from Ind48-CTRL simulations for the pre-monsoon season (a) latitude-pressure section (b) longitude-pressure section, (c)-(d) same as (a)-(b) but for the summer-monsoon season, (e)-(f) same as (a)-(b) but for the post-monsoon season, (g)-(h) same as (a)-(b) but for the winter season. The averages obtained over latitudes or longitudes are indicated in each panel. The black vertical bars indicate topography and a black line indicates the tropopause.




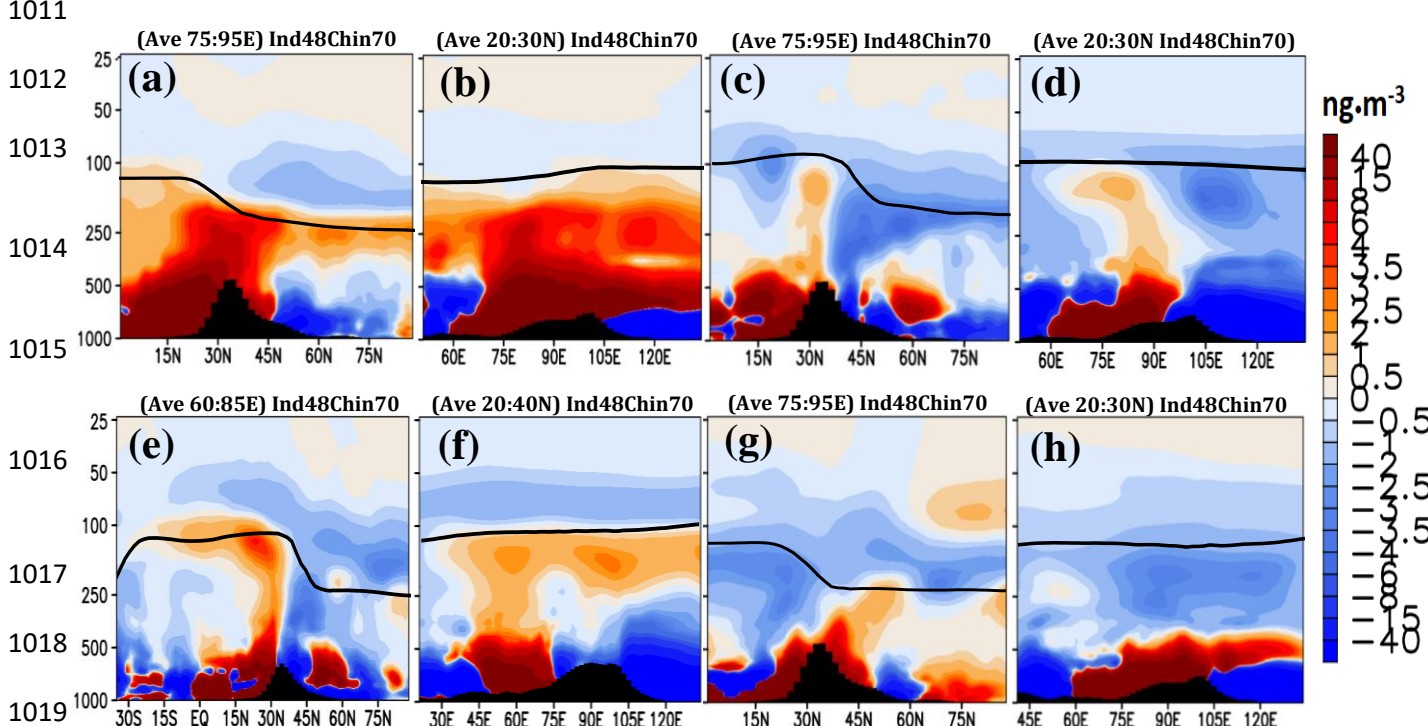

Figure 5: Vertical cross-section of anomalies in sulfate aerosols (ng·m$^{-3}$) from Ind48Chin70-CTRL simulation for the pre-monsoon season (a) latitude-pressure section (b) longitude-pressure section, (c)-(d) same as (a)-(b) but for the summer-monsoon season, (e)-(f) same as (a)-(b) but for the post-monsoon season, (g)-(h) same as (a)-(b) but for the winter season. The averages obtained over latitudes or longitudes are indicated in each panel. The black vertical bars indicate topography and a black line indicates the tropopause.

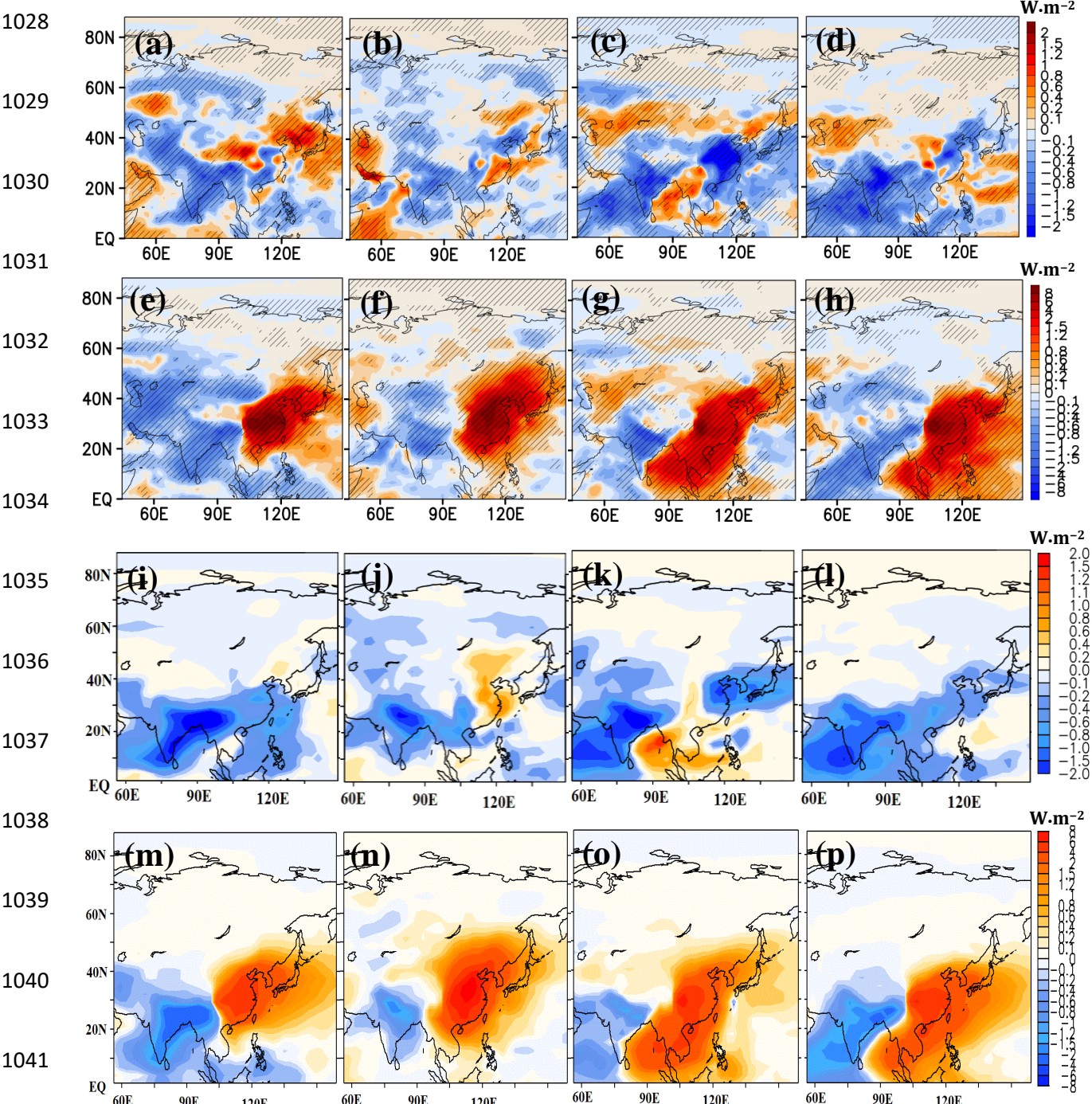

Figure 6: Seasonal distribution of anomalies in clear sky direct net radiative forcing (W·m$^{-2}$) simulated by ECHAM6–HAMMOZ at the top of the atmosphere, from Ind48-CRTL simulations for the (a) pre-monsoon (b) summer-monsoon, (c) post-monsoon and (d) winter season, (e)-(h) same as (a)-(d) but from Ind48Chin70-CTRL simulations. (i)-(l) same as (a)-(d) but from offline model, (m)-(p) same as (e)-(h) but from offline model. The black hatched lines in Figs. (a)-(h) indicate the 99 % significance level.

1048

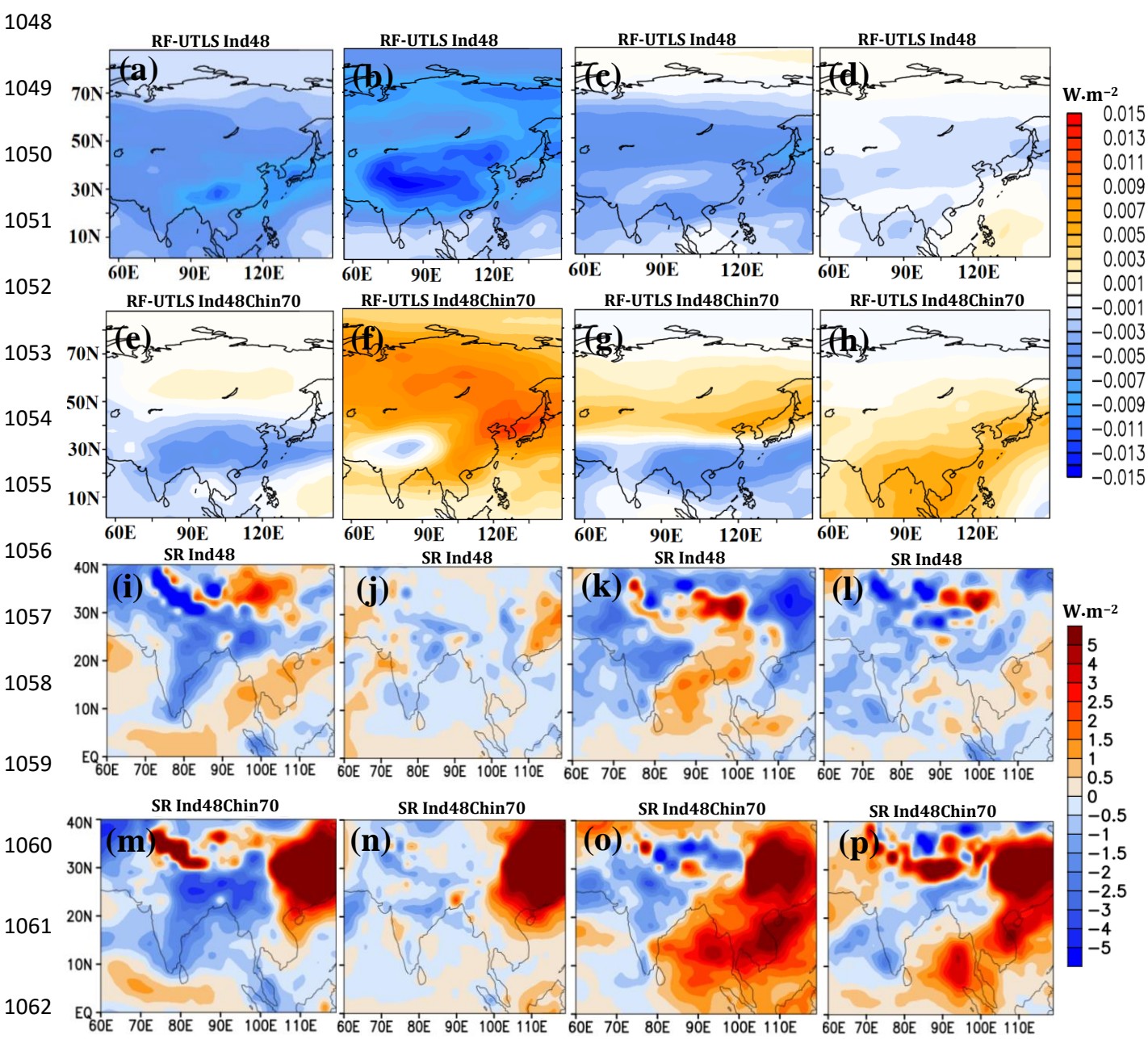

Figure 7: Simulated clear sky direct net radiative forcing at TOA (W·m⁻²) using the offline
model due to sulfate aerosols on the UTLS–only for the (a) pre-monsoon (b) summer-
monsoon, (c) post-monsoon, and (d) winter season for Ind48; (e)-(h) same as (a)-(d) but for
Ind48Chin70 simulations. Distribution of anomalies net solar radiation (SR) (W·m⁻²) at the
surface from Ind48 for the (i) pre-monsoon (j) summer-monsoon, (k) post-monsoon and (l)
winter season; (m)-(p) same as (i)-(l) but for Ind48Chin70 simulations.

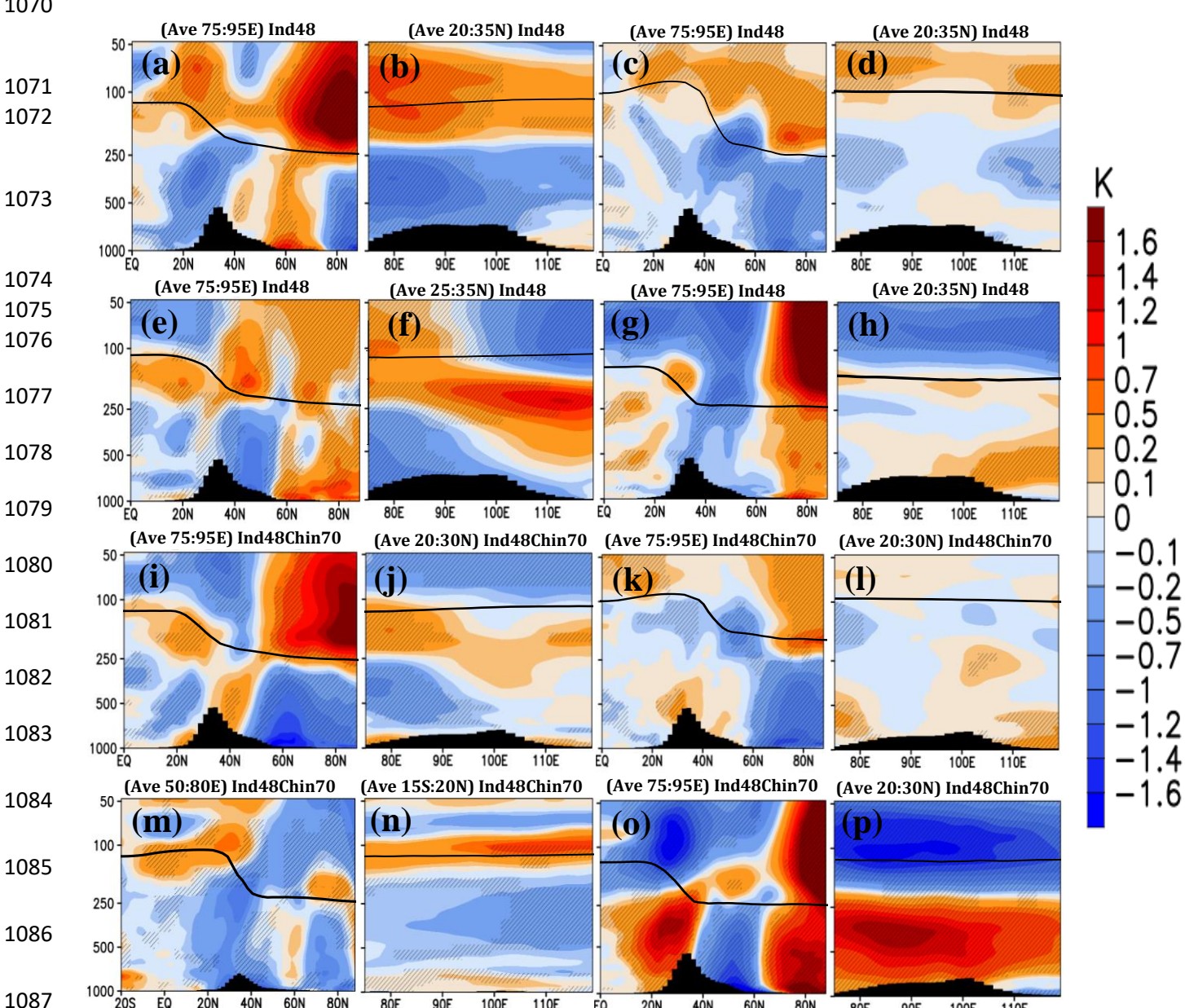

Figure 8: Vertical cross-section of anomalies in temperature (K) from Ind48-CRTL simulations for the pre-monsoon season (a) latitude-pressure section. (b) longitude-pressure section, (c)-(d) same as (a)-(b) but for the summer-monsoon season, (e)-(f) same as (a)-(b) but for the post-monsoon season, (g)-(h) same as (a)-(b) but for the winter season. Figures (i)-(p) same as (a)-(h) but from Ind48Chin70-CRTL simulations. For the vertical cross-section averages obtained over latitudes or longitudes are indicated in each panel. The black hatched lines indicate the 99 % significance level. The black vertical bars indicate topography and a black line indicates the tropopause.

1096

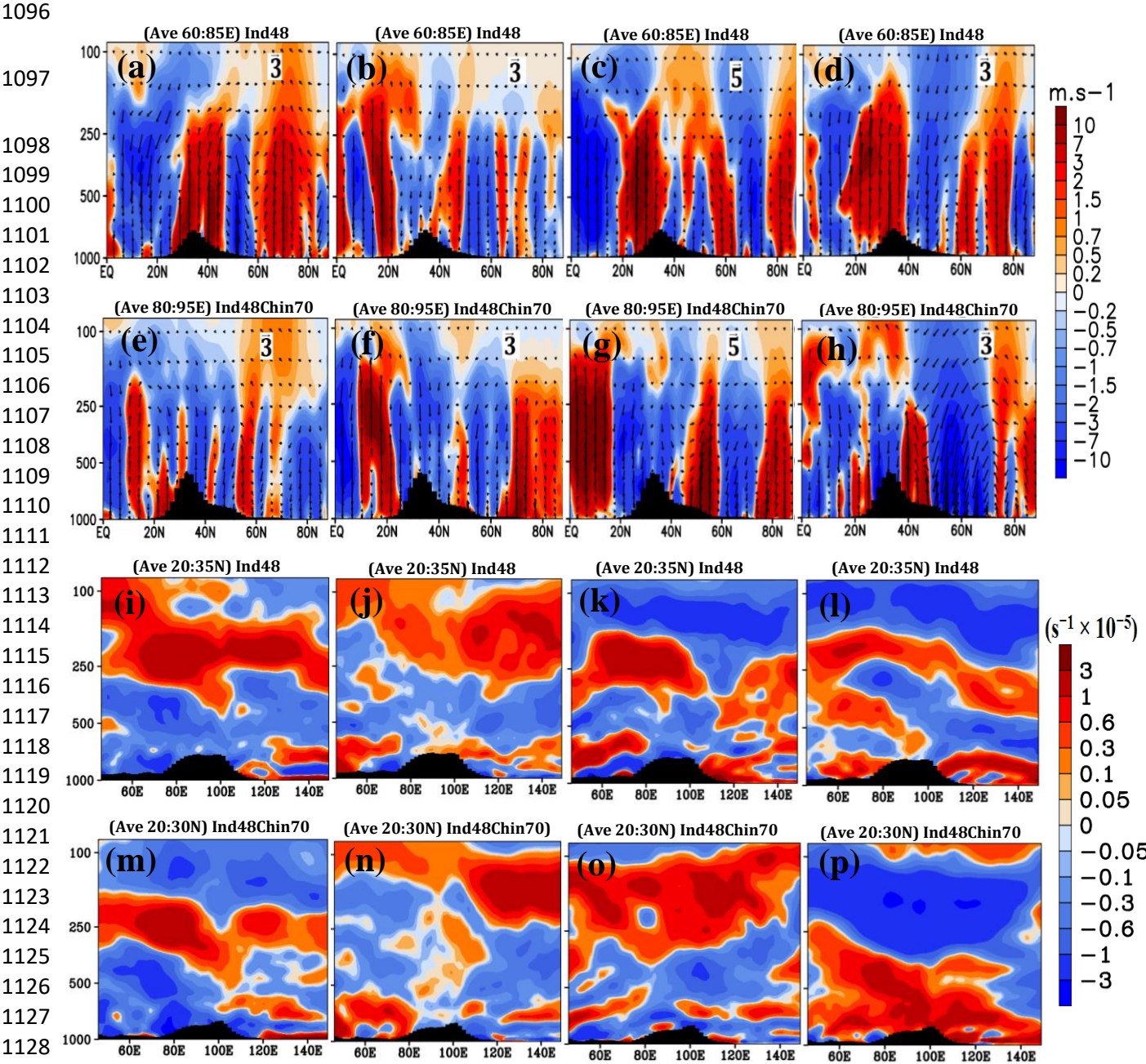

Figure 9: Distribution of anomalies in vertical velocity (m·s$^{-1}$) from Ind48-CTRLfor the (a)
pre-monsoon (b) summer-monsoon, (c) post-monsoon and (d) winter season, (e)-(h) same as
(a)-(d) but for Ind48Chin70-CTRL simulations. Vertical velocity is scaled by 1000. Seasonal
distribution of anomalies in Brunt–Väisälä frequency (s$^{-1}$ × 10$^{-5}$) from Ind48-CTRL for the (i)
pre-monsoon, (j) summer-monsoon, (k) post-monsoon and (l) winter season, (m)-(p) same as
(i)-(l) but from Ind48Chin70-CTRL simulations. For the vertical cross-section averages
obtained over latitudes or longitudes are indicated in each panel. The black vertical bars
indicate topography.

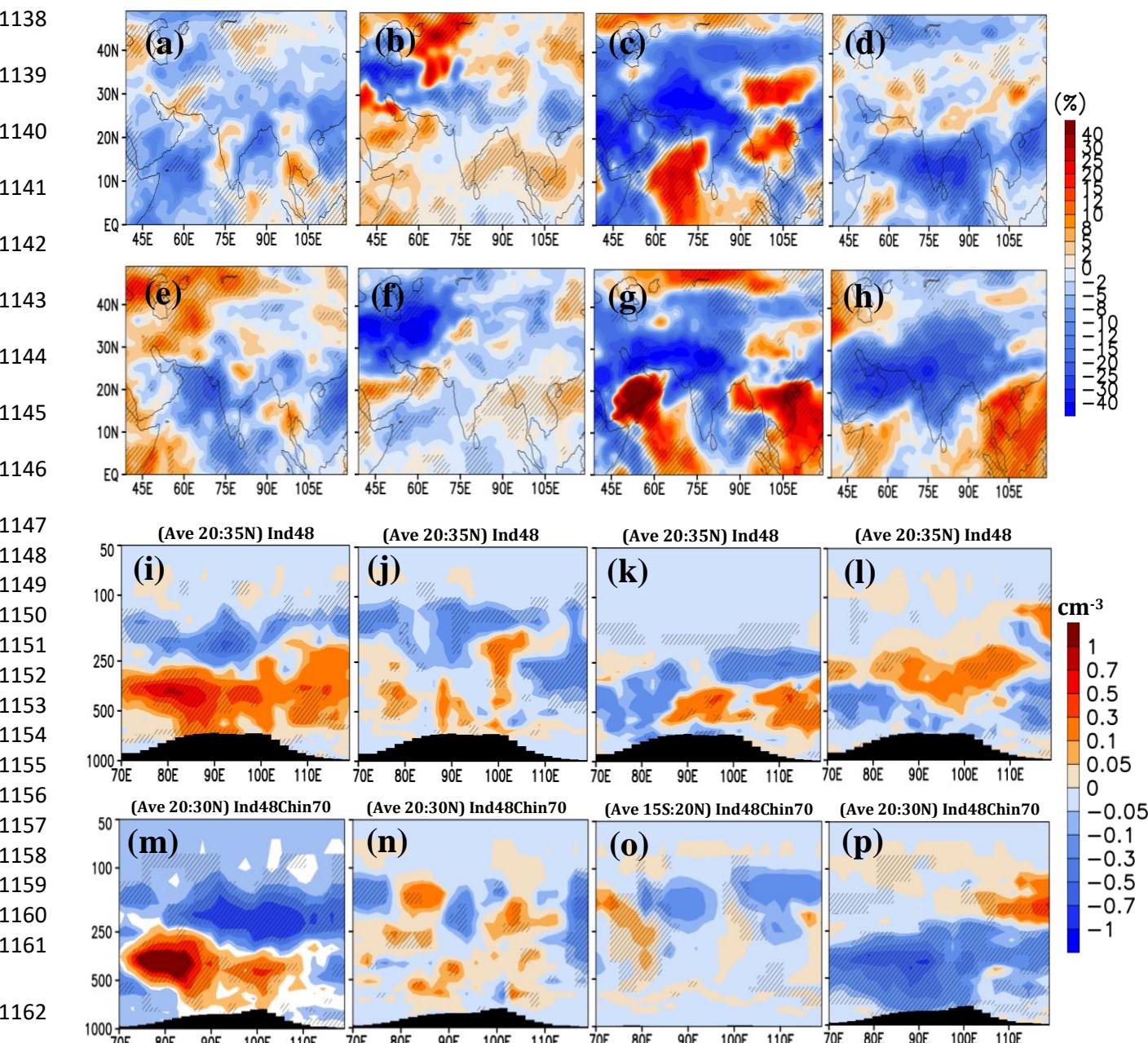

Figure 10: Seasonal distribution of anomalies in cirrus cloud (%) from Ind48-CRTL simulations for the (a) pre-monsoon, (b) summer-monsoon, (c) post-monsoon, and (d) winter season, (e)-(h) same as (a)-(d) but for Ind48Chin70-CTRL simulations, Seasonal distribution of anomalies in ICNC (cm$^{-3}$) from Ind48-CTRL for the (i) pre-monsoon, (j) summer-monsoon, (k) post-monsoon and (l) winter season, (m)-(p) same as (i)-(l) but from Ind48Chin70-CTRL simulations. For the vertical cross-section averages obtained over latitudes or longitudes are indicated in each panel. The black hatched lines indicate the 99 % significance level. The black vertical bars indicate topography.