# Peer review of "The impact of recent changes in Asian anthropogenic emissions of SO$_2$ on sulfate loading in the upper troposphere and lower stratosphere and the associated radiative changes"

_Atmospheric Chemistry and Physics, 2019_

## Referee Comment (RC1) · Anonymous Referee #2 · 2 Mar 2019

I commend the authors for their effort in addressing some issues with the original manuscript:

(1) regarding a possible explanation for the spatial maximum in transmitted solar radiation at the surface in west India at L389:

(2) for including AERONET AOD in this version of the manuscript.

(3) reducing the number of terms in the OMI $SO_2$ trend model

I have a few remaining major issues with this paper:

1) The AOD simulated by the ECHAM6-HAMMOZ climate-chemistry model does not agree well with that observed by MISR or AERONET (Fig. 1).

Could the model low bias in terms of AOD be due to a clear-sky bias of the AERONET and MISR observations? The authors could study this by including only times when there is not optically thick cloud in the time-average of the model AOD. Cloud information from the model could be used to filter the model AOD.

Because of this disagreement and the vertical transport of aerosols in the Asian monsoon to the UTLS, I think the authors need to have much larger uncertainties on the warming of the lower stratosphere by these (sulfate) aerosols: the abstract states that the warming reaches $0.6 \pm 0.25$ K. I would expect that the uncertainty on the warming should probably be comparable to the magnitude of the warming. From the uncertainty on the aerosol abundance available to reach the lower stratosphere (see Fig. 1e), I would expect a relative uncertainty of at least 75% on the warming (assuming a linear relationship between local aerosol concentration and local temperature).

The authors should remember that their warming estimate is obtained by the difference between two simulations (one being the control) and that differencing amplifies uncertainty if there is any kind of (numerical) noise in the model. Also, the method to determine the uncertainty of the warming estimates is not provided. The relative uncertainty on the $SO_2$ trend of 20% (= 0.97%/4.8%) could be added (in quadrature) to any other independent source of uncertainty.

2) The uncertainty on the $SO_2$ trend from OMI got smaller compared to the earlier version of the manuscript, i.e. $4.8 \pm 0.97$ %/yr versus $4.8 \pm 1.3$ %/yr. How did that happen? The uncertainty should have gotten larger with fewer terms.

3) The paper focussed on the monsoon season because convective transport in this season lifts boundary layer aerosols to the UTLS. But this is also a season with strong rainout and this might be leading to model bias since the rainout will be difficult to simulate accurately.

Furthermore, by limiting this paper to only the monsoon season, the conclusions are less interesting than a paper that providing the same analysis over a full year.

4) Overshooting convection (L313) is suggested as a pathway for aerosols to reach the stratosphere, but wouldn't the aerosols grow into cloud droplets with so much humidity and such strong convection? I suppose this is addressed by L75 of the introduction.

Why examine the $SO_2$ emissions from India only? The emissions from China have changed by the same magnitude but opposite sign (L83). The net effect might be very small if emissions from India and China were both perturbed according to their respective trends (based on OMI). Chinese emissions are very relevant to the Asian monsoon region.

Based on these issues (particularly #1), I feel the quality of this paper is somewhat low and the work is of limited scientific interest (see previous paragraph and issue #3). I do not find the model simulations convince me in terms of being realistic and yet the stated uncertainties are rather small. I suppose I could accept this paper if realistic and adequately described uncertainty calculations were included.

**Other general comments**

The model does not include nitrate, although it is acknowledged that it is an important aerosol species (L94,97,100,102, 258).

The reference list needs to improved. A consistent format is needed. Capital letters should not be used in common nouns in the article title, except for the first word of the title. See specific comments below.

**Specific comments**

L50: .The -> . The

L51: "Network of aerosol observatories ... Forcing (ARFINET)" -> "aerosol observatories ... Forcing Network (ARFINET)"

L73: Monsoon -> monsoon

L75: is -> are

L78: "pole ward" -> "poleward"

L86: "region" -> "monsoon (15-45°N, 30-120°E)"

L87: "tropical" -> "annually-averaged tropical (15°N-15°S)"

L95: "as a major ... component" -> "as major ... components"

L107: "forcing, for" -> "forcing. For"

L125: bases -> based

L192: OX -> Ox

L201: State the model's vertical resolution near the tropopause for a relevant latitude (i.e. 30°N).

L232: "satellite oservations" -> "observations via remote sensing"
    (to include AERONET)

L233 (and elsewhere): Use a comma after all leading prepositional phrases ("In Fig. 1a-b,"). See L208 for correct usage.

L257: Dumka et al., 2014 is cited but the reference list only has Dumka et al., 2010. I believe the statement regarding 50% of the aerosols being located above 4 km is not present in the 2010 paper. Also, this statement cannot be generally true; the fraction will be much less than 20% in polluted places and perhaps this value is appropriate for a remote observing location in the Himalayas where the surface is at an altitude of 2 km.

L264: If there are models that perform better than ECHAM6-HAMMOZ in terms of accurately simulating AOD, the authors should justify their decision to prefer ECHAM6-HAMMOZ.

L266: "High" -> "The large"

L276: most -> mostly

L277: thunderstorm -> thunderstorms

L290: I found this statement interesting. Good!

L298: I question whether deep convection (i.e. to the low-latitude upper troposphere) occurs over water (Bay of Bengal). Convective transport of relevance to the monsoon region is occurring in southeast Asia.

L306: aerosols -> aerosol

L318: "Sulfate" -> "The sulfate"

L329: "occurs on a daily scale" -> "it is of short duration (i.e., days) and is episodic".

L339-341: The authors speculate in many places in the paper (search for "may" or "likely"). I find it excessive and in this case, I wonder if the anomaly is significant (i.e., real).

L344: More speculation with little support... . An equally "likely" explanation in my opinion is the much greater concentration of sulfate in the lower troposphere between 50-70°N.

L349: move "~0.1 W m$^{-2}$" before "over" in L348.

L367: The thermal anomaly is really not that large. (-1 x 10$^{-3}$ K/day leads to a 0.1 K change after 100 days).

L374: "$CO_2$" -> "the $CO_2$"

L379 (Sect. 5.2): I believe the first paragraph here could be quite confusing for readers. The trend in radiative forcing from Ramanathan et al. is provided (but with the wrong units, should be W/m$^2$/yr) and then the next sentence presents the magnitude of the anomaly due to sulfur emissions found in this work. But the numbers are really not comparable since Ramanathan et al. were looking at a much longer and earlier period. I realize the authors may be trying to simply cite this related work here but I fear that readers will believe they should somehow compare the radiative forcing anomaly simulated in this work to the trend from Ramanathan et al. and/or Padma Kumari et al. This can be remedied by simply starting the sentence at L382 with "While not directly comparable to these previous studies, ... ".

L384: The authors are cherry-picking the evidence. Figure 6b does not show a very coherent pattern. While the tendency is for negative solar radiation at the surface in northern India, west India is not the only exception (e.g. east India).

L387: "connecting the boundary layer of the ASM region to the UTLS" sounds poetic, but it is not demonstrated in this paper. I believe the reduction in surface radiation is mainly due to aerosols in the boundary layer and the aerosols in the UTLS have a very minor contribution to the received shortwave radiation at the ground. This can be tested by removing the aerosols from the UTLS and looking at the change in shortwave radiation at the surface.

L389: "values of clouds" -> "cloud fractions"

L390: 5.1 -> 5.1.

L393, 395, 401, 402: I don't believe any of these uncertainties (i.e., too small).

L405: subsidence is not discussed in section 5.3. I suggest that "and subsidence" is removed here.

L412: Remove "the strong subsidence" or demonstrate it. This comment applies to L468 too.

L426: "liquid-origin history" -> "liquid origin"

L434: "anomalies are negative" does not belong in this sentence. Please reword so that this is a proper sentence.

L460: Re: "~-1.38", I question whether not only the "8" is a significant digit, but even the "3".

L462: There is not "good" agreement between the offline calculations and the model. Also, "minor" is absolutely not acceptable in the next sentence.

L547: "Beig,G." -> "Beig, G."          (there are spaces missing throughout the references, particularly in the author lists.

L565: indian -> Indian

L648: "PadmaKumari"  -> "Padma Kumari"

L659 (and elsewhere): 2018 -> 2018.

L667: (AIRS). -> (AIRS),

L672 (and elsewhere): "et al." is not acceptable for ACP last time I checked.

L686: "et al,." -> "et al.,"

L745: Do not use italics.

L758: "S, A." -> "S. A."

L746: Brenninkmeijer -> Brenninkmeijer,

L821 (Fig. 3): The black vertical bars are not described in the caption and should be removed because they block the colour contour plot.

L868: Net -> net

L869: radiations -> radiation

---

## Referee Comment (RC2) · Anonymous Referee #1 · 20 Mar 2019

Review of the paper: "The impact of increases in South Asian anthropogenic emissions of SO2 on sulfate loading in the upper troposphere and lower stratosphere during the monsoon season and the associated radiative impact", by S. Fadnavis et al., Atmos. Chem. Phys. Discuss., acp-2019-81, 2019.

This study focuses on the impact that rapidly increasing anthropogenic emissions of SO2 in South Asia may have on the distribution of UTLS sulfate. This is an important

topic and the manuscript deserves publication on ACP, after two major points (in my opinion) have been correctly addressed in the revised version.

Major points

1) The most important conclusions of the present study (changes in ATAL and related radiative forcing both at the surface and TOA, as well as feedback processes on UTLS dynamics and clouds) are based on the model calculated distribution of sulfate aerosols following the increasing anthropogenic SO2 emissions at the surface over South Asia. This distribution is not only determined by local convective uplift, but also by the lower stratospheric coupling of aerosol transport and microphysics. From this point of view, the quasi-biennal oscillation (QBO) plays a major role in determining the rate of large-scale isentropic transport from the tropics to the extratropics. A different SO2 and SO4 lifetime in the tropical reservoir may, in turn, affect the aerosol size distribution, thus modulating the sedimentation rate and the strat-trop exchange. Nothing is said in the manuscript on how the QBO is treated in the model simulations. Internally generated? External nudging? What is the different level of sulfate export from the tropical reservoir during E/W phase years? I think the authors should clarify and produce some evidence of the model predicted variability in the horizontal gradient of the sulfate loading between tropics and extratropics (maybe in the supplementary material). Some recent studies have focused on this topic, looking at model simulations for SO4 aerosols from sulfate geoengineering (e.g., Aquila et al., 2014; Visioni et al., 2018). It is true that in this latter case, as well as for aerosols from major tropical volcanic eruptions (e.g., Pinatubo; Trepte and Hitchman, 1992) the aerosols are located a few kilometrs above those convectively uplifted from the surface, but the QBO impact on the latitudinal transport of aerosols in the lower stratosphere should be significant, anyhow. The link between tropical UTLS sulfate (convectively uplifted from South Asia) and its poleward transport is mentioned in several places in the manuscript (lines 22-23, 77-79, 309-314, 321-325, 344-345, 398-399, 450-453) and is one of the key points in the discussion. For this reason, the QBO effects need to be addressed. 2) Proper

acknowledgment of previous works in the literature is needed. The authors cite the review paper of Kremser et al. (2016), but they should do the same for the SPARC assessment of stratospheric aerosol properties (ASAP, 2006), as well. Here, in the uncertainties section of Chapter 6, a detailed discussion is made on the potential impact of future trends of stratospheric sulfate aerosols due to increasing anthropogenic sulfur emission in South Asia. A citation to SPARC-ASAP would be appropriate, for example, at lines 76-77 and line 287.

Minor points

Both in the abstract (line 19) and in the conclusions (line 447) the authors write: "...experiments with SO2 emissions enhanced by 48% over South Asia...". For the reader, it is not clear (mainly in the abstract) with respect to what the emissions are enhanced by 48%. Later on in the text this is made clear (lines 210-213).

Line 66: "economy and agricolture" instead of "economy, agricolture".

Line 78 and 399: "poleward" is one word, not two.

A reference is missing at line 202: "AMIP (add reference) sea surface temperature...".

Line 340: is likely to be caused.

Lines 370-371: Ozone absorption of the increasing diffuse radiation by sulfate aerosols may also play a role.

At line 443 the Kuebbeler et al. (2012) citation is not appropriate for the cirrus cloud formation response to volcanic eruptions, but it should be moved at line 444 together with Visioni et al. (2018). On the other hand, the effects of non-explosive volcanic eruptions on UTLS aerosols and cirrus ice clouds were explored in Pitari et al. (2016).

References

Aquila et al.: Modifications of the quasi-biennial oscillation by a geoengineering perturbation of the stratospheric aerosol layer, Geophys. Res. Lett., 41, 1738–1744, 2014.

Visioni et al.: Sulfur deposition changes under sulfate geoengineering conditions: QBO effects on transport and lifetime of stratospheric aerosols, Atmos. Chem. Phys., 18, 2787-2808, doi: 10.5194/acp-18-2787-2018, 2018.

Pitari et al.: Sulfate aerosols from non-explosive volcanoes: chemical-radiative effects in the troposphere and lower stratosphere, Atmosphere, 7, 85, doi:10.3390/atmos7070085, 2016.

SPARC: Assessment of stratospheric aerosol properties (ASAP), L. Thomason and Th. Peter, Eds., www.sparc-climate.org, 2006.
* * *
* * *

---

## Author Comment (AC1) · 10 Jul 2019

Replies to Anonymous Referee #2

I commend the authors for their effort for addressing their issues with the original manuscript.

(1)regarding a possible explanation for the spatial maximum in transmitted solar radia-

tion at the surface in west India at L389 :

( 2 ) for including AERONET AOD in this version of the manuscript .

( 3 ) reducing the number of terms in the OMI SO2 trend model.

I have a few remaining major issues with this paper:

Reply: We thank the reviewer for the positive comments on our study and all valuable suggestions for improving the paper. In the revised version of the manuscript, we have addressed all the reviewer's comments through additional experiments and clarifying some key statements.

1 ) The AOD simulated by the ECHAM6–HAM-MOZ climate-chemistry model does not agree well with that observed by MISR or AERONET ( Fig .1). Could the model low bias in terms of AOD be due to a clear - sky bias of the AERONET and MISR observations? The authors could study this by including only times when there is not optically thick cloud in the time-average of the model AOD. Cloud information from the model could be used to filter the model AOD.

Reply(1): We agree this needs to be clarified in the manuscript. We have now added a discussion providing reasons for the regional underestimation/overestimation of AOD in the model (L277-282). AOD is calculated for clear-sky conditions, based on the assumption that the cloudy fraction of a grid-box is at saturation. The remaining available humidity is accessible for aerosol growth in clear-skies.

We have discussed the procedure for including AOD only times when there is not an optically thick cloud in the model with the ECHAM6-HAMOZ model development team. The team argues that while it is not optimal, a more careful collocation of observations and model data often underestimates AOD compared to satellite observations (for example MISR) (mainly over land; over the ocean there are large areas where the model actually overestimates).

Most atmospheric chemistry-climate models do not agree well with satellite remote

sensing observations. In the past, several papers have been published stating that majority of CMIP5 models underestimate AOD due to several reasons (Sanap et al., 2014; IPCC; 2013; Sockol et al., 2017) e.g., dust optical depth is underestimated in the model (Pu and Ginoux, 2018). There are uncertainties in model estimates of sea salt emission and parameterization (Spada et al., 2013).

We now include additional details in the revised manuscript at L279-282.

Sockol, A., and J. D. Small Griswold (2017), Inter-comparison between CMIP5 model and MODIS satellite-retrieved data of aerosol optical depth, cloud fraction, and cloud-aerosol interactions, Earth and Space Science, 4, 485–505, doi:10.1002/2017EA000288.

Sanap et al., (2014), Assessment of the aerosol distribution over Indian subcontinent in CMIP5 models, Atmospheric Environment 87, 123–137, DOI: 10.1016/j.atmosenv.2014.01.017.

(2) Because of this disagreement and the vertical transport of aerosols in the Asian monsoon to the UTLS, I think the authors need to have much larger uncertainties on the warming of the lower stratosphere by these (sulfate) aerosols: the abstract states that the warming reaches $0.6 \pm 0.25$ K. I would expect that the uncertainty on the warming should probably be comparable to the magnitude of the warming. From the uncertainty on the aerosol abundance available to reach the lower stratosphere (see Fig. 1e), I would expect a relative uncertainty of at least 75 % on the warming (assuming a linear relationship between local aerosol concentration and local temperature) The authors should remember that their warming estimate is obtained by the difference between two simulations (one being the control) and that differencing amplifies uncertainty if there is any kind of (numerical) noise in the model. Also, the method to determine the uncertainty of the warming estimates is not provided. The relative uncertainty on the SO2 trend of 20 % ( = $0.97$ % / $4.8$ % ) could be added ( in quadrature ) to any other independent source of uncertainty.

Reply(2): Yes, as mentioned we have obtained warming estimates from the difference between the two simulations. Each of the CTRL, Ind48 and Ind48Chin70 simulations are ensemble mean of 10 members and deviations within 10 members is represented as uncertainty. As the number of members increases, this uncertainty reduces. In the past, to obtain statistically robust results ten member simulations are performed by the numbers of researchers e.g., Lau et al., (2017); Jin et al., (2016); Fadnavis et al., (2013).

Following Lau et al., 2017; Jin et al., 2016; Fadnavis et al., 2013, we have adopted the approach of 10 member ensemble simulation for the year 2011. Also, confidence intervals are now replaced with 99% interval.

The uncertainty in SO2 trends obtained from OMI data is also given at a 99% confidence interval level. It is mentioned at L166.

Lau W. K-M. et al., Impacts of aerosol–monsoon interaction on rainfall and circulation over Northern India and the Himalaya Foothills, Climate Dynamics, 49, 1945–1960, 2017.

Jin Q., Yang Z-L, and Wei J., Seasonal Responses of Indian Summer Monsoon to Dust Aerosols in the Middle East, India, and China, J.Clim, 29, 6329-6349, 2016.

Fadnavis, S., et al.,: Transport of aerosols into the UTLS and their impact on the Asian monsoon region as seen in a global model simulation, Atmos. Chem. Phys., 13, 8771–8786, https://doi.org/10.5194/acp-13-8771-2013, 2013.

3 ) The uncertainty on the SO2 trend from OMI got small compared to the earlier version of the manuscript, i .e . 4 .8 $\pm$ 0 .9 7 % / yr versus 4.8 $\pm$ 1 .3 % / y r. How did that happen? The uncertainty should have gotten larger with fewer terms.

Reply(3): Although terms were reduced in equation 2 the level of confidence was also reduced. Now the uncertainty in SO2 trends obtained from OMI data is also given at a 99 % confidence interval level. Therefore, now it is given as 4.8 $\pm$ 3.2 % yr-1 over

India and 7.0 $\pm$ 6.3 % yr-1 over China. It is mentioned at L166.

4) The paper focussed on the monsoon season because convective transport in this season lifts boundary layer aerosols to the UTLS. But this is also a season with a strong rainout, and this might be leading to model bias since the rain out will be difficult to simulate accurately. Furthermore, by limiting this paper to only the monsoon season, the conclusions are less interesting than a paper that providing the same analysis over a full year.

Reply(4): As suggested we have provided analysis for a full year from additional simulations for India and China SO2 emissions perturbed according to the respective trends based in OMI observations during 2006 – 2017, (1) SO2 emission over India increased by 48 %, (2) SO2 emission over India increased by 48 % (Ind48) and decreased over China by 70 % simultaneously (Ind48Chin70).

5) Over shooting convection (L313) is suggested as a pathway for aerosols to reach the stratosphere, but would't the aerosols grow in to cloud droplets with so much humidity and such strong convection? I suppose this is addressed by L75 of the introduction. Reply(5): It is now mentioned at L86.

6) Why examine the SO2 emissions from India only? The emissions from China have changed by the same magnitude but opposite sign (L83). The net effect might be very small if emissions from India and China were both perturbed according to the respective trends (based on OMI). Chinese emissions are very relevant to the Asian monsoon region. Based on these issues (particularly # 1), I feel the quality of this paper is some what low and the work is of limited scientific interest (see the previous paragraph and issue # 3). I do not find the model simulations convince me in terms of being realistic and yet the stated uncertainties are rather small. I suppose I could accept this paper if realistic and adequately described uncertainty calculations were included.

Reply(6): As mentioned in reply(3) we have now included results from additional sim-

ulations where SO2 emissions over India and China are perturbed according to the trends observed by the OMI satellite.

The uncertainty estimates are replaced by 99 % confidence interval levels.

As mentioned in reply 1-3, we have shown uncertainties within 10 members of simulations.

Other general comments

(7) The model does not include nitrate, although it is acknowledged that it is an important aerosol species (L94 , 97, 100, 102, 258 ) .

Reply(7): The uncertainty due to absence of nitrate aerosol is mentioned in the manuscript at L277 as "Inclusion of nitrate aerosol may affect the distribution of the AOD."

(8) The reference list needs to be improved. A consistent format is needed. Capital letters should not be used in common nouns in the article title, except for the first word of the title.

Reply(8): As suggested reference list is improved and capital letters are removed from the common nouns in the article title.

See specific comments below.

(9) Specific comments L5 0 : . The - > .Th e

Reply(9): It is corrected at L59 . (10) L51 : " Network of aerosol observatories . . . Forcing ( ARFINET ) " - > " aerosol observatories . . . Forcing Network (ARFINET ) "

Reply (10): The long form of ARFINET is 'Aerosol Radiative Forcing over India NETwork (ARFINET)' is corrected at L59-60.

(13) L 7 3 : Monsoon - > monsoon

Reply(13) : The sentence is re-framed now.

(14) L7 5 : is - > are

Reply(14) : the above sentence is reframed.

(15) L78 : " poleward " - > " poleward "

Reply(15):. It is corrected at L66, L87, L320, L359, L379, L380, L397, L473.

(16) L 8 6 : " region " - > " monsoon ( 15 - 45 N , 30 - 120 E ) "

Reply(16) : It is corrected at L92.

(17) L 8 7 : " tropical " - > " annually - averaged tropical ( 15 N - 15 S ) " Reply(17): It is corrected at L94.

(18) L 9 5 : " as a major . . . component " - > " as major . . . components "

Reply(18): It is corrected at L102.

(19) L107 : " forcing , for " - > " forcing . For "

Reply(19) : It is corrected at L110.

(20) L125 : bases - > based

Reply(20): It is corrected at L138.

(21) L 1 9 2 : OX - > O x

Reply(21) : It is corrected at L201.

(22) L 2 0 1 : State the model ' s vertical resolution near the tropopause for a relevant latitude ( i .e . 30 N ) .

Reply(22) : It is corrected at L209 - L210.

(23) L 2 3 2 : " satellite observations " - > " observations via remote sensing " (to include AERONET)
[Figure]

Reply(23) : It is corrected at L247.

(24) L 2 3 3 ( and elsewhere ) : Use a comma after all leading prepositional phrases ( " In Fig. 1a - b , " ) . See L208 for correct usage .

Reply(24) : It is corrected at L248.

(25) L257 : Dumka et al . , 2014 is cited but the reference list only has Dumka et al., 2010. I believe the statement regarding g 50 % of the aerosols being located above 4 k m is not present in the 2010 paper. Also, this statement cannot be generally true ; the fraction will be much less than 20 % in polluted places and perhaps this value is appropriate for a remote observing location in the Himalayas where the surface is at an altitude of 2 k m .

Reply(25) : The above sentence is re-phrased as: Dumka et al. (2014) has documented that in AERONET observations the aerosols above 4 km contribute 50 % of AOD at Kanpur (in Indo-Gangetic plains). See the snapshot below:

(26) L 2 6 4 : If there are models that perform better than ECHAM6 - HAMMOZ in terms of accurately simulating A O D , the authors should justify their decision to prefer ECHAM6 - HAMMOZ .

Reply(26) : As mentioned in reply(1), we have added discussion and reasons for regional under-estimation or overestimation of AOD in the model (L275-282). The model underestimates AOD over India while it overestimates over China in comparison with MISR. While it is underestimates over both the regions in comparison with AERONET.

(27) L 2 6 6 : " High " - > " The large "

Reply(27) : It is corrected at L281.

(28) L 2 7 6 : most - > mostly

Reply(28) : It removed in the revised manuscript.

(29) L 2 7 7 : thunderstorm - > thunderstorms

Reply(29) : It is now removed from the revised manuscript.

(30) L 2 9 0: I found this statement interesting . Good !

Reply (30): Thank you.

(31) L 2 9 8: I question whether deep convection ( i e . to the low - latitude upper troposphere ) occurs over water ( Bay of Bengal ). Convective transport of relevance to the monsoon region is occurring in South East Asia.

Reply(31): The trajectory-based analysis and other modelling studies show that deep convection occurring over the Bay of Bengal, southern slopes of Himalayas and the South China Sea play an important role in convective transport of South Asian pollution to the upper troposphere. The moist, cloud-laden air lifted to the upper troposphere by convective systems from these regions (Chen et al., ACP, 2012; Fadnavis et al., 2013; Dong et al., 2016). The forced lifting of air by high orography of Himalayas plays a vital role in transport into the UTLS. Convective systems at the Himalayas are associated with Bay of Bengal depressions, as strong low-level flow transports maritime moisture into the region (Medina et al., 2010).

The above discussion is pertaining to the monsoon season while the current manuscript discusses all the seasons. Therefore it is now removed from the manuscript. We have added supplementary figure 1 depicting regions of convection during different seasons using OLR from NCEP; Ice-crystal-number concentration and cloud-droplet number concentration from ECHAM6-HAMMOZ.

Medina S., Houze Jr R. A., Kumar A., Niyogi D., Summer monsoon convection in the Himalayan region: Terrain and land cover effects, Quarterly Journal of the Royal Meteorological Society 136(648):593 – 616,DOI: 10.1002/qj.601, 2010.

(32) L 3 0 6 :aerosols - > aerosol

Reply(32): It is corrected at L319.

(33) L 3 1 8 : " Sulfate " - > " The sulfate "

Reply(33): The sentence is removed from the revised manuscript.

(34) L 3 2 9 : " occurs on a daily scale " - > " it is of short duration ( i . e . , days) and is episodic.

Reply (34): It is corrected at L348-L349.

(35):L 3 3 9 - 3 4 1 : The authors speculate in many places in the paper ( search for " may " or " likely " ) . I find it excessive and in this case, I wonder if the anomaly is significant ( ie . , real ) . L 3 4 4 : More speculation with little support . . . . An equally " likely " explanation, in my opinion, is the much greater concentration of sulfate in the lower troposphere between 50 - 70  N .

Reply(35): Now, above sentences are written as affirmative.

(36)L 3 4 9: move " $\sim 0 . 1$ W m - 2 " before " over " in L348 .

Reply(36): The above sentence is reframed.

(37) L 3 6 7: The thermal anomaly is really not that large. (- 1 x 1 0 -3 K / day leads to a 0 . 1 K change after 100 days ).

Reply(37): It is removed from the revised manuscript.

(38) L 3 7 4 : " CO2 " - > " the CO2 "

Reply(38) : It is corrected at L445.

(39) L 3 7 9 ( S e c t . 5 . 2 ): I believe the first paragraph here could be quite confusing for readers. The trend in radiative forcing from Ramanathan et al. is provided ( but with the wrong units, should be W/m2/y r ) and then the next sentence presents the magnitude of the anomaly due to sulfur emissions found in this work. But the numbers are really not comparable since Ramanathan et al. were looking at a much longer

and earlier period. I realize the authors may be trying to simply cite this related work here but I fear that readers will be believe they should somehow compare the radiative forcing anomaly simulated in this work to the trend from Ramanathan et al . and / or Padma Kumari et al . This can be remedied by simply starting the sentenceat L382 with " While not directly comparable to these previous studies , . . . " .

Reply (39): As suggested we have added the line "While not directly comparable to these previous studies" at L 450-451.

(40) L 384: The authors are cherry - picking the evidence . Figure 6 b does not show a very coherent pattern . While the tendency is for negative solar radiation at the surface in northern India , west India is not the only exception ( e . g . east India ) .

Reply (40): We agree that this needs to be clarified and thatnk the reviewer for pointing this out. The above sentence is re-written at L455-463 as "We estimate the changes in net solar radiation at the surface for four seasons from the Ind48 and Ind48Chin70 simulations. Figure 7i-l shows that the Ind48 simulations have produced negative anomalies in net solar radiation (SR) at the surface ($\sim$-0.5 to -3 W•m$-$2) over India and parts of China (where sulfate aerosols are transported) due to the enhanced sulfate aerosol layer reflecting back solar radiation. In general, the seasonal mean distribution of anomalies in net solar radiation at the surface is similar to the distribution of the anomalies in RF at the TOA. Reduction of Chinese SO2 emissions along with an increase of SO2 emissions over India (Ind48Chin70) has produced a reduction of solar radiation over India while there is a significant increase over China (1 $-$ 5 W•m$-$2) (see Fig 7 m-p)".

(41) L 3 8 7: " connecting the boundary layer of the ASM region to the UTLS " sounds poetic, but it is not demonstrated in this paper. I believe the reduction in surface radiation is mainly due to aerosols in the boundary layer and the aerosols in the UTLS have a very minor contribution to the received short wave radiation at the ground. This can be tested by removing the aerosols from the UTLS and looking at the change in short

wave radiation at the surface.

Reply(41) : The above sentence is removed.

(42) L 3 8 9 : " values of clouds " - > " cloud fractions " L 3 9 0 : 5 . 1 - >5 .1 . L 3 9 3 , 3 9 5 , 4 0 1 , 4 0 2 : I don 't believe any of these uncertainties ( i . e . , too small) .

Reply (42) : We have now given uncertainties at a 99 % confidence level.

(43) L 4 0 5 : subsidence is not discussed in section 5 . 3. I suggest that "and subsidence" is removed here.

Reply(43) : As suggested it is removed.

(44) L 4 1 2 :Remove " the strong subsidence " or demonstrate it . This comment applies to L468 too.

Reply(44) : We have removed 'the strong subsidence" and re-written it as 'upper tropospheric cooling and enhanced stability may suppress the rainfall' at L501-502.

(45) L 4 2 6 : " liquid – origin history " - > " liquid origin "

Reply (45) : The above correction is incorporated at L515.

(46) L 4 3 4 : " anomalies are negative " does not belong in this sentence. Please reword so that this is a proper sentence.

Reply (46): It is re-written as "Figure 10 a-h shows the impact of SO2 emission changes on cirrus clouds. It shows a decrease (5 – 30 %) of cirrus clouds over North India (20 – 35 °N) in the UTLS" (L520-524).

(47) L 460 : Re : " $\sim$ - 1 . 3 8 " , I question whether not only the " 8 " is a significant digit , but even the " 3 " .

Reply( 47): The above reframed now at L583.

(48) L 4 6 2 : There is not " good " agreement between the off-line calculations and the

model. Also, " minor " is absolutely not acceptable in the next sentence .

Reply (48): This sentence is re-written as "These values are comparable with results of the ECHAM6–HAMMOZ simulations, with the minor differences likely due to the implicit dynamical impacts in response to enhanced south Asian $SO_2$ emissions in ECHAM6–HAMMOZ not being represented in the offline model". L584-587.

(49) L 5 4 7 : " Beig , G. " - > " Beig , G." (there are spaces missing throughout the references , particularly in the author lists) .

Reply(49): It is corrected at L687.

(50)L 5 6 5 :indian - > Indian

Reply(50): It is removed in the revised version.

(51) L 6 48 : " PadmaKumari " - > " Padma Kumari "

Reply(51) : It is corrected at L796.

(52) L 6 5 9 (and else where ) : 2018 - > 2018 .

Reply(52) : It is corrected now.

(53) L 6 6 7 : ( AIRS ) . - >( AIRS ) ,

Reply(53) : It is removed in the revised version.

(54) L 6 7 2 ( and else where ) : " et al . " is not acceptable for ACP last time I checked .

Reply(54) : It is corrected now.

(55) L 6 8 6 : " e t a l , . " - >" e t a l . , "

Reply(55) : It is corrected at L836.

(56) L 7 4 5 : Do not use italics . L 7 5 8 : " S , A . " - >" S . A . "

Reply(56) : It is removed in the revised version.

(57) L 7 4 6 :Brenninkmeijer - >Brenninkmeijer ,

Reply (57) : It is removed in the revised version.

(58) L 8 2 1 ( F i g . 3 ) : The black vertical bars are not described in the caption and should be removed because they block the colour contour plot .

Reply(58): The back vertical bars indicate topography. It is mentioned in the figures caption (Fig, 4,5,9,10)

(59)L 8 6 8 : Net - > net

Reply(59): It is corrected at L1040.

(60) L869 : radiations - >radiation

Reply(60) : It is corrected at L1064.

---

## Author Comment (AC2) · 10 Jul 2019

Replies to Anonymous Referee #1

Review of the paper: "The impact of increases in South Asian anthropogenic emissions of SO2 on sulfate loading in the upper troposphere and lower stratosphere during the monsoon season and the associated radiative impact", by S. Fadnavis et al., Atmos. Chem. Phys. Discuss., acp-2019-81, 2019. This study focuses on the impact that

rapidly increasing anthropogenic emissions of SO2 in South Asia may have on the distribution of UTLS sulfate. This is an important topic and the manuscript deserves publication on ACP, after two major points (in my opinion) have been correctly addressed in the revised version.

Reply: We thank the reviewer for the positive comments on our paper and all valuable suggestions. We have performed additional experiments and tried to incorporate suggestions given by the reviewer.

Major points (1) The most important conclusions of the present study (changes in ATAL and related radiative forcing both at the surface and TOA, as well as feedback processes on UTLS dynamics and clouds) are based on the model calculated distribution of sulfate aerosols following the increasing anthropogenic SO2 emissions at the surface over South Asia. This distribution is not only determined by local convective uplift, but also by the lower stratospheric coupling of aerosol transport and microphysics. From this point of view, the quasi-biennal oscillation (QBO) plays a major role in determining the rate of large-scale isentropic transport from the tropics to the extra-tropics. A different SO2 and SO4 lifetime in the tropical reservoir may, in turn, affect the aerosol size distribution, thus modulating the sedimentation rate and the strat-trop exchange. Nothing is said in the manuscript on how the QBO is treated in the model simulations. Internally generated? External nudging? What is the different level of sulfate export from the tropical reservoir during E/W phase years? I think the authors should clarify and produce some evidence of the model predicted variability in the horizontal gradient of the sulfate loading between tropics and extra-tropics (maybe in the supplementary material).

Reply(1):. We agree these are important points that need to be clarified. The focus of our manuscript is to understand the convective transport of Asian sulfate aerosols during the monsoon season. Therefore free simulations (10 member ensemble) were

performed for the year 2011 with a one-year spin-up for the year 2010. The analysis is presented for the year 2011. These experiments are canonical in design as their aim is to understand the radiative impact of Asian sulfate aerosols. The model results do not include the influence of QBO, which has a periodicity of 22-24 months. Also, the QBO is not internally generated in the model. We now clarify this in the manuscript at L231-232.

We thank the reviewer for the valuable suggestion about analyzing the role of the quasi-biennial oscillation (QBO) in understanding the large-scale isentropic transport from the tropics to the extra-tropics. QBO can be generated in the model by the external nudging. To understand the influence of enhancement of sulfate aerosols on QBO, we have now performed external nudging experiments for the years 2008 - 2016 (CTRL and Ind48 simulations). Our model simulations show that the enhancement of sulfate aerosols slows down the QBO propagation (Figure 1a-b below). There is interannual variability in transport sulfate aerosols by the phases of QBO (Figure 1c). It affects the transport out of the tropics (Figure 1d). Since the focus of the present paper is to highlight the seasonal transport and associated radiative impacts, we plan to provide detail analysis of the interaction of QBO and sulfate aerosol in a separate paper which will focus on "Influence of sulfate aerosol on QBO: implications on Asian summer monsoon convection".

Following the reviewer's suggestions, we have now added a discussion about sulfate export from the tropical reservoir during E/W phase of QBO (section 6 in the manuscript).

(2) Some recent studies have focused on this topic, looking at model simulations for SO4 aerosols from sulfate geoengineering (e.g., Aquila et al., 2014; Visioni et al., 2018). It is true that in this latter case, as well as for aerosols from major tropical volcanic eruptions (e.g., Pinatubo; Trepte and Hitchman, 1992) the aerosols are located a few kilometers above those convectively uplifted from the surface, but the QBO impact on the latitudinal transport of aerosols in the lower stratosphere should be significant, anyhow. The link between tropical UTLS sulfate (convectively uplifted from South Asia) and its poleward transport is mentioned in several places in the manuscript (lines 22-23, 77-79, 309-314, 321-325, 344-345, 398-399, 450-453) and is one of the key points in the discussion. For this reason, need to be addressed.

Reply(2): Thank you for this important point. As mentioned in the reply(1), the QBO is not internally generated in the model (L231-232). To see the influence of QBO on the transport in our model, nudge simulations need to be performed. However, the model simulations used in the manuscript are 10-member ensemble free runs. These runs show poleward transport of the Asian sulfate aerosols by the lowermost branch of the Brewer-Dobson circulation.

We have also added a discussion about sulfate export from the tropical reservoir during East-West phases in discussion section 6.

3) Proper acknowledgment of previous works in the literature is needed. The authors cite the review paper of Kremser et al. (2016), but they should do the same for the SPARC assessment of stratospheric aerosol properties (ASAP, 2006), as well. Here, in the uncertainties section of Chapter 6, a detailed discussion is made on the potential impact of future trends of stratospheric sulfate aerosols due to increasing anthropogenic sulfur emission in South Asia. A citation to SPARC-ASAP would be appropriate, for example, at lines 76-77 and line 287.

Reply(3) : We agree. As suggested citation of SPARC-ASAP is added at L81 and L315.

4) Minor points Both in the abstract (line 19) and in the conclusions (line 447) the authors write: ". . .experiments with SO2 emissions enhanced by 48% over South Asia. . .". For the reader, it is not clear (mainly in the abstract) with respect to what the emissions are enhanced by 48%. Later on in the text this is made clear (lines 210-213).

Reply(4) : As suggested, SO2 emissions enhancement by 48% over South Asia in the model experiment is made clear in abstract and conclusion (L18-21 and L567-570).

(5) Line 66: "economy and agriculture" instead of "economy, agriculture".

Reply(5) : The above-said sentence is removed from the manuscript.

(6) Line 78 and 399: "poleward" is one word, not two.

Reply(6) : Above mentioned suggestion is incorporated at L66, L87, L320, L359, L379, L380, L397, L473.

(7) A reference is missing at line 202: "AMIP (add reference) sea surface temperature".

Reply(7) : We have incorporated the reference (Taylor et al., 2002) at L211.

(8) Line 340: is likely to be caused.

Reply(8) : Above mentioned phrase is not used in revised manuscript.

(9) Lines 370-371: Ozone absorption of the increasing diffuse radiation by sulfate aerosols may also play a role.

Reply(9): The above sentence is now removed from the revised manuscript.

(10) At line 443 the Kuebbeler et al. (2012) citation is not appropriate for the cirrus cloud formation response to volcanic eruptions, but it should be moved at line 444 together with Visioni et al. (2018). On the other hand, the effects of non-explosive volcanic eruptions on UTLS aerosols and cirrus ice clouds were explored in Pitari et al. (2016).

Reply(10): The citations of Kuebbeler et al. (2012) and Visioni et al. (2018) are moved to L535-536. Pitari et al. (2016) is added at L536.

(11) References Aquila et al.: Modifications of the quasi-biennial oscillation by a geo-engineering perturbation of the stratospheric aerosol layer, Geophys.

Reply(11) : The above reference is added at L546.

(12) Visioni et al.: Sulfur deposition changes under sulfate geoengineering conditions: QBO effects on transport and lifetime of stratospheric aerosols, Atmos. Chem. Phys., 18, 2787-2808, doi: 10.5194/acp-18-2787-2018, 2018.

Reply(12) : The above reference is added at L547.

(13) Pitari et al.: Sulfate aerosols from non-explosive volcanoes: chemical radiative effects in the troposphere and lower stratosphere, Atmosphere, 7, 85, doi:10.3390/atmos7070085, 2016.

Reply(13) : The above reference is added at L536, L493.

(14) SPARC: Assessment of stratospheric aerosol properties (ASAP), L. Thomason, and Th. Peter, Eds., www.sparc-climate.org, 2006.

Reply(14) : The above reference is added at L315, L834.

Please also note the supplement to this comment:
https://www.atmos-chem-phys-discuss.net/acp-2019-81/acp-2019-81-AC2-supplement.pdf

**Supplement:**

[revised manuscript text omitted]

---

## Author Comment (AC3) · 10 Jul 2019

(1) The most important conclusions of the present study (changes in ATAL and related radiative forcing both at the surface and TOA, as well as feedback processes on UTLS dynamics and clouds) are based on the model calculated distribution of sulfate aerosols following the increasing anthropogenic SO2 emissions at the surface over South Asia. This distribution is not only determined by local convective uplift, but also

by the lower stratospheric coupling of aerosol transport and microphysics. From this point of view, the quasi-biennal oscillation (QBO) plays a major role in determining the rate of large-scale isentropic transport from the tropics to the extra-tropics. A different SO2 and SO4 lifetime in the tropical reservoir may, in turn, affect the aerosol size distribution, thus modulating the sedimentation rate and the strat-trop exchange. Nothing is said in the manuscript on how the QBO is treated in the model simulations. Internally generated? External nudging? What is the different level of sulfate export from the tropical reservoir during E/W phase years? I think the authors should clarify and produce some evidence of the model predicted variability in the horizontal gradient of the sulfate loading between tropics and extra-tropics (maybe in the supplementary material).

Reply(1):. We agree these are important points that need to be clarified. The focus of our manuscript is to understand the convective transport of Asian sulfate aerosols during the monsoon season. Therefore free simulations (10 member ensemble) were performed for the year 2011 with a one-year spin-up for the year 2010. The analysis is presented for the year 2011. These experiments are canonical in design as their aim is to understand the radiative impact of Asian sulfate aerosols. The model results do not include the influence of QBO, which has a periodicity of 22-24 months. Also, the QBO is not internally generated in the model. We now clarify this in the manuscript at L236.

We thank the reviewer for the valuable suggestion about analyzing the role of the quasi-biennial oscillation (QBO) in understanding the large-scale isentropic transport from the tropics to the extra-tropics. QBO can be generated in the model by the external nudging. To understand the influence of enhancement of sulfate aerosols on QBO, we have now performed external nudging experiments for the years 2008 - 2016 (CTRL and Ind48 simulations). Our model simulations show that the enhancement of sulfate aerosols slows down the QBO propagation (Figure 1a-b below). There is interannual variability in transport sulfate aerosols by the phases of QBO (Figure 1c). It affects the transport out of the tropics (Figure 1d). Since the focus of the present paper is to highlight the seasonal transport and associated radiative impacts, we plan to provide detail analysis of the interaction of QBO and sulfate aerosol in a separate paper which will focus on "Influence of sulfate aerosol on QBO: implications on Asian summer monsoon convection".

Following the reviewer's suggestions, we have now added a discussion about sulfate export from the tropical reservoir during E/W phase of QBO (section 6 in the manuscript).

See below the attached Figure-1

[Figure]

[Figure]

Figure-1: ECHAM6-HAMMOZ simulated vertical distribution of zonal winds (2008 -
2015) averaged for 5 ºS – 5 ºN (a) CTRL, (b) Ind48 simulations,(c) vertical distribution of
anomalies of sulfate aerosols (Ind48-CTRL) over North India (70 - 95ºE; 20 - 40ºN)
during 2008 – 2015, (d) anomalies of sulfate aerosols (Ind48-CTRL) averaged over 70 -
95º E for the year 2011. Arrows in Fig.1(d) indicate the transport of sulfate aerosols from
North India.

**Fig. 1.**

---

## Referee Report (RR1)

Once again, I am encouraged by the effort of the authors. They have extended the study to a whole year, which is more relevant for climate.

L165: In the authors' response, the authors mention that the uncertainty they are quoting is based on the 99% confidence interval. But the revised manuscript contains the newly inserted: "significant at 99% level". Is this related or are the authors trying to express that the trend is positive and significant? If the authors are simply stating the confidence interval (C. I.), the text should read "(99% confidence interval)". It is fine to state there is an increasing trend that is statistically significant but the reader needs to know what the uncertainty represents. So, if 3.2% is not the 99% C. I., then what is this uncertainty?

L477: It should be explicitly stated that the temperature uncertainties in this paragraph are simply obtained by determining the variability within the 10-member ensemble (if I am correctly understanding how the uncertainties were determined).

L1007: back -> black

---

## Author Response (AR2)

**Replies to the Editor**

Before publication in ACP there are some minor issues that should be considered as given in the referee report. Additionally, I would like to ask you to consider the following technical corrections

Reply: We hank the editor for appreciating our efforts. We have incorporated the suggestion given by the Editor. The changes are indicated in blue colour at line numbers indicated in the replies below.

P5, L101: is, ......., as a major -> something is wrong here. Please correct. I guess "as" is obsolete.

Reply: It is corrected at L101.

P6, L107: gas-to-aerosol -> gas-to-particle

Reply: It is corrected at L107.

P6, L114: Add a space after full stop.

Reply: It is corrected at L114.

P16, L331: ....the Arctic maximizing during ......-> that maximizes during........

Reply: It is corrected at L330-331.

P16, L336: twice region -> please rephrase

Reply: It is rephrased as "from the west Asia and Tibetan Plateau region (20 – 35 °N; 60 – 95 °E). This may be due to the transport of sulfate aerosols from India to these regions, which might have been lifted to the UTLS by the post-monsoon convection (see Figs. S1 c, h, k, and S2 c) at L334-337.

P24, L524: "is" obsolete

Reply: It is corrected at L525.

P25, L548: "it" obsolete

Reply: It is corrected at L549.

P25, L550: add "a" so that it reads "for a larger amount".

Reply: It is corrected at L551.

P26, L570: Ind48 simulations show -> The Ind48 simulations shows

Reply: It is corrected at L571.

P27, L588: with sulfate aerosol layer -> with the aerosol sulfate layer

Reply: It is corrected at L588-589.

P28, L600: is inhibited the vertical transport -> please rephrase

Reply: It is rephrased as 'Reduction of Chinese $SO_2$ emissions does not stabilize the upper troposphere during monsoon and winter seasons since subsidence over North India inhibited the vertical transport of sulfate aerosols to the UTLS' at L598-601.

**Replies to the Reviewer-II**

Once again, I am encouraged by the effort of the authors. They have extended the study to a whole year, which is more relevant for climate.

Reply: We thank the reviewer for valuable suggestions and appreciating our efforts. We have included suggestions given by the reviewer. The changes are indicated in blue colour in the manuscript and corresponding line numbers are indicated in each of the reply below.

L165: In the authors' response, the authors mention that the uncertainty they are quoting is based on the 99% confidence interval. But the revised manuscript contains the newly inserted: "significant at 99% level". Is this related or are the authors trying to express that the trend is positive and significant? If the authors are simply stating the confidence interval (C. I.), the text should read "(99% confidence interval)". It is fine to state there is an increasing trend that is statistically significant but the reader needs to know what the uncertainty represents. So, if 3.2% is not the 99% C. I., then what is this uncertainty?

Reply: The above sentence is rephrased as '99% confidence interval' at L165.

L477: It should be explicitly stated that the temperature uncertainties in this paragraph are simply obtained by determining the variability within the 10-member ensemble (if I am correctly understanding how the uncertainties were determined).

Reply: It is stated at L479-480.

L1007: back -> black

Reply: It is corrected at L1008.